

# Transport of Po Valley aerosol pollution to the northwestern Alps – Part 2: Long-term impact on air quality

Henri Diémoz[1], Gian Paolo Gobbi[2], Tiziana Magri[1], Giordano Pession[1], Sara Pittavino[1], Ivan K. F. Tombolato[1], Monica Campanelli[2], and Francesca Barnaba[2]

[1]ARPA Valle d'Aosta, Saint-Christophe, Italy
[2]Institute of Atmospheric Science and Climate, CNR, Rome, Italy

*Correspondence to:* Henri Diémoz (h.diemoz@arpa.vda.it)

**Abstract.**

This work evaluates the impact of trans-regional aerosol transport from the polluted Po basin on particulate matter levels ($PM_{10}$) and physico-chemical characteristics in the northwestern Alps. To this purpose, we exploited a multi-sensor, multi-platform database over a 3-years period (2015–2017) accompanied by a series of numerical simulations. The experimental

setup included operational (24/7) vertically-resolved aerosol profiles by an Automated LiDAR-Ceilometer (ALC), vertically-integrated aerosol properties by a sun/sky photometer, and surface measurements of aerosol mass concentration, size distribution and chemical composition. This experimental set of observations was then complemented by modelling tools, including Numerical Weather Prediction (NWP), Trajectory Statistical (TSM) and Chemical Transport (CTM) models, plus Positive Matrix Factorisation (PMF) on both the $PM_{10}$ chemical speciation analyses and size distributions. In a first companion study

(Diémoz et al., 2019), we showed and discussed through detailed case studies the 4-D phenomenology of recurrent episodes of aerosol transport from the polluted Po basin to the northwestern Italian Alps, and particularly to the Aosta Valley. Here we draw more general and statistically significant conclusions on the frequency of occurrence of this phenomenon, and on the quantitative impact of this regular, wind-driven, aerosol-rich "atmospheric tide" on $PM_{10}$ air quality levels in this alpine environment. Combining vertically-resolved ALC measurements with wind information, we found that an advected aerosol

layer is observed at the receptor site (Aosta) in 93% of days characterized by easterly winds (thermally-driven winds from the plain or synoptic circulation regimes), and that the longer the time spent by air masses over the Po plain the higher this probability. On a seasonal basis, frequency of advected aerosol layers from the Po basin maximises in summer (70% of the days classified using the ALC profiles) and minimises in winter and spring (57% of the classified days). Duration of these advection events ranges from few hours up to several days, while aerosol layer thickness ranges from 500 up to 4000 m. This

phenomenon was found to largely impact both surface levels and column-integrated aerosol properties, with $PM_{10}$ and AOD values respectively increasing up to a factor of 3.5 and 4 in dates under the Po Valley influence. Similar variations in $PM_{10}$ values observed at different stations within the Aosta Valley also indicated the phenomenon to act at the regional scale and to be related to non-local emissions. Pollution transport events were also shown to modify the mean chemical composition and typical size of particles in the target region. In fact, increase in secondary species, and mainly nitrate- and sulfate-rich

components, were found to be effective proxies of the advections, with the transported aerosol responsible for at least 25% of





the $PM_{10}$ measured in the urban site of Aosta, and adding up to over $50\,\mu g\,m^{-3}$ during specific episodes, thus exceeding alone the EU established daily limit. This percentage is expected to be higher in the rural, pristine areas on the northwestern Alps, where chemical data were not available and trans-boundary contribution to $PM_{10}$ might thus exceed the local one. Advected aerosols were also found to be on average finer, more light-scattering and more hygroscopic than the locally-produced ones.

From a modelling point of view, our CTM simulations performed over a full year showed that the model is able to reproduce the phenomenon but underestimates its impact on $PM_{10}$ levels. As a sensitivity test, we employed the ALC-derived identification of aerosol advections to re-weight the emissions from outside the boundaries of the regional domain in order to match the observed $PM_{10}$ field. This simplified exercise indicated that an increase of such "external" emissions by a factor of 4 in the model would reduce the $PM_{10}$ mean bias forecasts error (MBE) from $-10\,\mu g\,m^{-3}$ to less than $2\,\mu g\,m^{-3}$, the normalised mean standard deviation (NMSD) from over $-50\%$ to less than $-10\%$ and would halve the model $PM_{10}$ maximum deviations.

# 1 Introduction

Mountain regions are often considered pristine areas, being typically far from large urban settlements and strong anthropogenic emission sources, and thus relatively unaffected by remarkable pollution footprints. However, atmospheric transport of pollutants from the neighbouring foreland is not uncommon, owing to the synoptic and regional circulation patterns. For example, in several regions of the world, thermally-driven flows represent a systematic characteristic of mountain weather and climate (Hann, 1879; Thyer, 1966; Kastendeuch and Kaufmann, 1997; Egger et al., 2000; Ying et al., 2009; Serafin and Zardi, 2010; Schmidli, 2013; Zardi and Whiteman, 2013; Wagner et al., 2014; Giovannini et al., 2017; Schmidli et al., 2018) favouring regular exchange of air masses between the plain and the highland sites (Weissmann et al., 2005; Gohm et al., 2009; Cong et al., 2015; Dhungel et al., 2018), with likely consequences on human health (Loomis et al., 2013; EEA, 2015; WHO, 2016), ecosystems (e.g., Carslaw et al., 2010; EMEP, 2016; Bourgeois et al., 2018; Burkhardt et al., 2018), climate (Ramanathan et al., 2001; Clerici and Mélin, 2008; Philipona, 2013; Pepin et al., 2015; Zeng et al., 2015; Tudoroiu et al., 2016; Samset, 2018) and, not least, local economy, through loss of tourism revenues (de Freitas, 2003; Keiser et al., 2018). These phenomena have worldwide relevance, since nearly one quarter of the Earth's land mass can be classified as mountainous areas (Blyth, 2002).

Among them, the Alps are of particular interest, due to both their sensitive environment and their proximity to the Po Valley (Fig. 1a; Tampieri et al., 1981; Seibert et al., 1998; Wotawa et al., 2000; Dosio et al., 2002; Campana et al., 2005; Kaiser, 2009; Finardi et al., 2014). In fact, the Po basin represents a major pollution hotspot in Europe, with large emissions from highly populated urban areas (about 40% of the Italian population lives in this region, WMO, 2012), intense anthropogenic activities, such as industry and agriculture, combined to a local topography promoting atmospheric stability (Chu et al., 2003; Barnaba and Gobbi, 2004; Schaap et al., 2004; Van Donkelaar et al., 2010; Fuzzi et al., 2015; Bigi and Ghermandi, 2014; EMEP, 2016; Belis et al., 2017; EEA, 2017). Despite the efforts to decrease the number of particulate matter (PM) exceedances, Italy is still failing to comply with the European air quality standards (EU Commission, 2008, 2018).

In a first companion paper (Diémoz et al., 2019), we investigated the phenomenology of the aerosol advections from the Po basin to the northwestern Alps, and specifically to the Aosta Valley (Fig. 1b). This was introduced and thoroughly described by





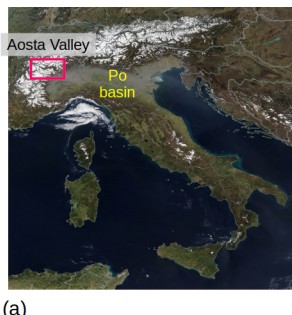

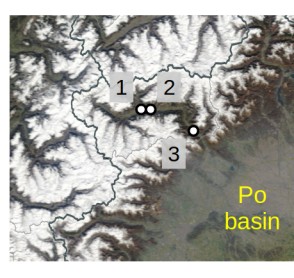

(a)  (b)

**Figure 1.** True colour corrected reflectance from MODIS Terra satellite (http://worldview.earthdata.nasa.gov) on 17 March 2017. (a) Italy, with indication of the Aosta Valley (northwest side, rectangle) and the Po Valley. (b) Zoom over the Aosta Valley. The circle markers represent the sites of 1) Aosta–Downtown; 2) Aosta–Saint Christophe; 3) Donnas. A thin aerosol layer over the Po basin starting to spread out into the Alpine valleys is visible in both figures.

means of three specifically-selected case studies, each of them lasting several days and monitored by a large set of instruments. That investigation evidenced clear features of this phenomenology, and notably: 1) the aerosol transported to the northwestern Alps is clearly detectable and discernible from the locally-produced aerosol. Detection of such polluted aerosol layers over Aosta was primarily driven by remote-sensing profiling measurements (Automated LiDAR Ceilometers, ALCs), these show-
ing recurrent arrival of thick and elevated aerosol-rich layers. Good correspondence was found between the presence of these layers aloft, changes of column-averaged aerosol properties, and variations of PM surface concentrations and chemical composition; 2) the advected particles are small (accumulation mode), mainly of secondary origin, weakly light-absorbing and highly hygroscopic compared to the locally-produced aerosol. Clouds are frequently observed to form within these layers during the night; 3) the air masses associated to the observed elevated layers are found to originate from the Po basin and to be trans-
ported to the northwestern Alps by up-valley breezes and synoptic winds; 4) the chemical transport model (FARM, Flexible Air quality Regional Model) currently used by the local Environment Protection Agency (ARPA Valle d'Aosta) was able to reproduce this transport from a qualitative point of view and was usefully employed to interpret the observations. However, it fails at quantitatively estimating the PM mass contribution from outside the boundaries of the regional domain (approximately corresponding to the Valle d'Aosta region), likely due to deficiencies in the emission inventories and to unaccounted aerosol
processes in the model (the main of which are likely those related to hygroscopicity/aqueous-phase chemistry triggered by high relative humidities).

The present work analyses the same phenomenon, but with a long-term perspective. Main aim of this study is to establish frequency of occurrence of aerosol transport from the Po Valley to the northwestern Alps and its impact on local air quality. In particular, this paper addresses the following questions:

1. How frequently does advection from the Po plain to the northwestern Alps occur? What are the most common meteorological conditions favouring it?



2. What are the average properties of the advected aerosol both at the surface and in the layers aloft?

3. How much does transport from the Po basin impact the air quality metrics in the northwestern Alps?

4. Can we effectively predict the arrival of aerosol-polluted air masses to the northwestern Alps? How can we improve our air quality forecasts?

To answer these questions, we took advantage of long-term (2015–2017) and almost uninterrupted series of measurements. The experimental dataset, thoroughly described by Diémoz et al. (2019), included measurements of vertically-resolved aerosol profiles by an ALC, vertically-integrated aerosol properties by a sun/sky photometer and surface measurements of the aerosol mass concentration, size distribution and chemical composition, most of these series encompassing a time period of several years. Indeed, a great number of intensive, short-term campaigns employing cutting-edge research instruments and techniques were already performed on this topic in the Po basin, mainly focussing on its central, eastern and southern parts (Nyeki et al., 2002; Barnaba et al., 2007; Ferrero et al., 2010, 2014; Saarikoski et al., 2012; Landi et al., 2013; Decesari et al., 2014; Costabile et al., 2017; Bucci et al., 2018; Cugerone et al., 2018). However, continuous and multi-year datasets, especially from ground-based stations, are necessary to assess the influence of pollution transport on a longer term (e.g., Mélin and Zibordi, 2005; Clerici and Mélin, 2008; Kambezidis and Kaskaoutis, 2008; Mazzola et al., 2010; Bigi and Ghermandi, 2014, 2016; Putaud et al., 2014; Arvani et al., 2016). In this context, local environmental agencies operate stable networks for continuous monitoring of air quality and meteorological parameters, and use standardised methodologies and universally recognised quality control procedures.

The study description is organised as follows: the investigated area is presented in Sect. 2, together with the experimental setup, further complemented by mathematical tools (Sect. 3). Results are presented in Sect. 4 and include a) a first, original classification framework based on ALC data and meteorological variables; b) a long-term, statistical analysis of the phenomenon and its impacts on surface/column aerosol properties and c) comparison between observations and CTM simulations. Conclusions are drawn in Sect. 5.

## 2 Study region and experimental setup

We provide here a brief overview of the region of interest and of the experimental setup, this latter being described with full details in the companion paper (Diémoz et al., 2019).

The Aosta Valley, located at the northwestern Italian border (Fig. 1), is the smallest administrative region of the country, being only 80 km by 40 km wide and hosting 130000 inhabitants. The most remarkable geographical feature of the region, whose mean altitude exceeds 2000 m a.s.l., is the main valley, approximately oriented in a SE to NW direction and connecting the Po basin (about 300 m a.s.l, at the southeast side of the Aosta Valley region) to the Mont Blanc chain (4810 m a.s.l., separating Italy and France). Several tributary valleys depart from the main valley. The topography of the area triggers some of the most common weather regimes in the mountains, such as thermally-driven winds (from the east, during daytime), slope winds perpendicular to and along the main valley axis, rain-shadow winds ("Foehn", from the west), and frequent temperature





inversions during wintertime anticyclonic days. Quite obviously, those meteorological phenomena are responsible of most of the variability of the atmospheric constituents (e.g., water vapour, Campanelli et al. (2018), and pollutants concentrations, Agnesod et al. (2003)).

The air quality network of the local Environmental Protection Agency is designed trying to capture the topographical and meteorological heterogeneity of the investigated area. In this study, we mainly used data from three sites belonging to the network (Fig. 1b): Aosta–Saint Christophe (560 m a.s.l., semi-rural), Aosta–Downtown (580 m a.s.l., urban background) and Donnas (316 m a.s.l., rural). The first two measuring stations, located 2.5 km apart, are situated respectively in the suburbs and in the centre of Aosta (45.7°N, 7.4°E), the main urban settlement of the region (36000 inhabitants). Car traffic, domestic heating and a steel mill are the main anthropogenic emission sources within the city, whose pollution levels are generally moderate (e.g., yearly average $PM_{10}$ concentration of about 20 $\mu g\,m^{-3}$, with summertime and wintertime averages of 13 and 31 $\mu g\,m^{-3}$, respectively). Donnas is located at the border with the Po basin and is expected to be strongly impacted by pollution from outside the region. In fact, the $PM_{10}$ concentration in this rural site is about 18 $\mu g\,m^{-3}$ on a yearly average, i.e. only slightly lower than the concentration in Aosta.

The Aosta–Saint Christophe atmospheric observatory (WIGOS ID 0-380-5-1, Diémoz et al., 2011, 2014a) is located on the terrace of the ARPA building. At this site, advanced instrumentation is continuously operated, daily checked and maintained. Vertically-resolved measurements of aerosols and clouds are performed by a commercially-available Automated LiDAR Ceilometer (ALC, Lufft CHM15k-Nimbus) running since 2015 in the framework of the Italian Alice-net (http://www.alice-net.eu/) and the European E-PROFILE (http://ceilometer.e-profile.eu/profileview) networks. The ALC emits in the atmosphere 8 μJ laser impulses at a single wavelength of 1064 nm with a frequency of 7 kHz and collects their backscatter by aerosol and molecules from ground up to an altitude of 15 km, with a vertical resolution of 15 m. This return signal is averaged over 15 s and saved by the firmware as range-, overlap- and baseline-corrected raw counts ($\beta_{raw}$). To convert the backscatter signal in SI units, a Rayleigh calibration (Fernald, 1984; Klett, 1985; Wiegner and Geiß, 2012) is necessary. This is accomplished by the operator based on data sampled during clear-sky nights. This allows to derive the particle backscatter coefficient ($\beta_p$) and the scattering ratio (SR, e.g. Zuev et al., 2017), which is the first, ALC-derived parameter used in this work to quantify the aerosol load:

$$SR = \frac{\beta_p + \beta_m}{\beta_m} \qquad (1)$$

$\beta_m$ being the molecular backscattering coefficient (thus SR=1 indicates a purely molecular atmosphere while SR increases with aerosol content).

At the same station, a POM-02 sun/sky photometer operates since 2012 as part of the European ESR-SKYNET network (http://www.euroskyrad.net/). The instrument collects the direct light coming from the sun (every 1 min) and the light scattered from the sky in the almucantar plane (every 10 min) to retrieve the aerosol optical depth (AOD) and other aerosol optical and microphysical properties at 11 wavelengths (315-2200 nm) using the SKYRAD.pack version 4 code. The sun photometer is calibrated in situ with the improved Langley technique (Campanelli et al., 2007) and was recently successfully compared





to other reference instruments (Kazadzis et al., 2018). Accurate cloud screening is achieved using the Cloud Screening of Sky Radiometer data (CSSR) algorithm by Khatri and Takamura (2009). Special care is needed to retrieve, among the other parameters, the aerosol single scattering albedo (SSA). In fact, it is known that the SSA retrieval by the SKYRAD.pack version 4 is problematic and sometimes unnaturally close to unity (Hashimoto et al., 2012), irrespectively of the AOD. Although

improvements are expected in the next versions of SKYRAD.pack, data collected within the European ESR-SKYNET network are still processed with version 4. Therefore, only SSA lower than 0.99 at all wavelengths were retained in our analysis.

Finally, a Fidas 200s Optical Particle Counter (OPC) is used in Aosta–Saint Christophe to yield an accurate estimation of the aerosol size distribution (0.18–18 μm) in proximity of the surface (Pletscher et al., 2016). This instrument, although not directly measuring the aerosol mass, obtained the certificate of equivalence to the gravimetric method by TÜV Rheinland Energy GmbH

on the basis of a laboratory and a field test. A $PM_{10}$ comparison with the gravimetric technique was additionally performed in Aosta–Saint Christophe and provided satisfactory results (29 days; slope $1.08\pm0.04$; intercept $-3.8\pm1.4\,\mathrm{\mu g\,m^{-3}}$; $R^2 = 0.96$). Further instruments to monitor trace gases (Diémoz et al., 2014b) are employed at the station and the corresponding datasets will be investigated in future studies.

Aosta–Downtown hosts a monitoring station for continuous measurements of the aerosol concentration and composition.

$PM_{10}$ and $PM_{2.5}$ daily concentrations are available from two Opsis SM200 Particulate Monitor instruments. An estimate of the $PM_{10}$ hourly variability is furthermore provided by a Tapered Element Oscillating Microbalance (TEOM) 1400a monitor (Patashnick and Rupprecht, 1991), though not compensated for mass loss of semi-volatile compounds (Green et al., 2009), such as ammonium nitrate (e.g., Charron et al., 2004; Rosati et al., 2016). Moreover, the collected $PM_{10}$ samples are chemically analysed on a daily basis. Ion Chromatography (AQUION/ICS-1000 modules) is employed for determining the concentration

of water-soluble anions and cations ($Cl^-$, $NO_3^-$, $SO_4^{2-}$, $Na^+$, $NH_4^+$, $K^+$, $Mg^{2+}$, $Ca^{2+}$), based on the CEN/TR 16269:2011 guideline. Elemental/organic carbon (EC/OC), using the thermal-optical transmission (TOT) method (Birch and Cary, 1996) and following the EUSAAR-2 protocol (Cavalli et al., 2010) according to the EN 16909:2017, are analysed alternatively to metal concentrations (Cr, Cu, Fe, Mn, Ni, Pb, Zn, As, Cd, Mo and Co), by means of inductively coupled plasma mass spectrometry (ICP-MS) after acid mineralisation of the filter in aqueous solution (EN 14902:2005). Particle-bound polycyclic aromatic

hydrocarbons (PAHs) are detected continously in real-time by a photoelectric aerosol sensor (EcoChem PAS-2000). Gas-phase pollutants, such as NO and $NO_2$ are also continuously monitored and were used in the present work to help identification of urban combustion sources (mainly traffic and residential heating).

Some of the air quality parameters monitored in Aosta–Downtown are also measured at the Donnas station, i.e. $PM_{10}$ daily concentrations (Opsis SM200) and nitrogen oxides (Horiba APNA-370 chemiluminescence monitor). Finally, the three stations

are equipped with instruments for tracking the standard meteorological parameters, such as temperature, pressure, relative humidity (RH) and surface wind velocity. A list of the instruments, with relevant operating period and subset considered in this study, is presented in Table 1.



**Table 1.** Observation sites, measurements and instruments employed in this study.

| Station | Elevation (m a.s.l.) | Measurement | Instrument | Data availability | Used in this study |
|---|---|---|---|---|---|
| Aosta–Saint Christophe (ARPA observatory) | 560 | Vertical profile of attenuated backscatter and derived products | CHM15k-Nimbus ceilometer | 2015–now | 2015–2017 |
| | | Aerosol columnar properties | POM-02 sun/sky radiometer | 2012–now[a] | 2015–2017 |
| | | Surface particle size distribution | Fidas 200s optical particle counter | 2016–now | 2016–2017 |
| Aosta–Saint Christophe (weather station) | 545 | Standard meteorological parameters | Micros (wind) | 1974–now | 2015–2017 |
| Aosta–Downtown | 580 | $PM_{10}$ hourly concentration | TEOM 1400a | 1997–now | 2015–2017 |
| | | $PM_{10}$ and $PM_{2.5}$ daily concentrations | Opsis SM200 | 2011–now | 2015–2017 |
| | | Water-soluble anion/cation analyses on $PM_{10}$ samples | Dionex Ion Chromatography System | 2017–now | 2017 |
| | | EC/OC analyses on $PM_{10}$ samples | Sunset thermo-optical analyser | 2017–now[b] | 2017 |
| | | Metals on $PM_{10}$ samples | Varian 820 MS | 2000–now[c] | 2017 |
| | | Total PAHs | EcoChem PAS-2000 | 1995–now | 2017 |
| | | NO and NO2 | Horiba APNA-370 | 1995–now | 2017 |
| | | Standard meteorological parameters | Vaisala WA15 (wind) | 1995–now | 2015–2017 |
| Donnas | 316 | $PM_{10}$ daily concentration | Opsis SM200 | 2010–now | 2015–2017 |
| | | NO and NO2 | Teledyne API200E | 1995–now | 2015–2017 |

[a] Underwent major maintenance in the second half of 2016 and January 2017.

[b] Available 4 days every 10.

[c] Available 6 days every 10.

## 3 Modelling tools

Numerical models and statistical techniques were used to interpret and complement the observations described above. A numerical weather prediction model (COSMO, Consortium for Small-scale Modeling, www.cosmo-model.org) was employed to drive a chemical transport model (FARM, Flexible Air quality Regional Model) and a lagrangian model (LAGRANTO). These





tools are only briefly recalled here (Sect. 3.1), since they were thoroughly described by Diémoz et al. (2019). Conversely, Trajectory Statistical Models (TSMs) and Positive Matrix Factorisation (PMF) were adopted to interpret the long-term series of measurements in this work, and are therefore fully described in Sects. 3.2 and 3.3, respectively.

### 3.1 Numerical atmospheric models

COSMO is a non-hydrostatic, fully compressible atmospheric prediction model working on the meso-$\beta$ and meso-$\gamma$ scales (Baldauf et al., 2011). Owing to the complex topography of the Aosta Valley, and the consequent need to resolve the atmospheric circulation at very small spatial scales, in the present work we used a nudged, high-resolution (2.8 km, 65 vertical levels, 2 runs/day) variant, called COSMO-I2 (or COSMO-IT), covering Italy. The COSMO forecasts, with the complete set of parameters (including the 3-D wind velocity used here), are disseminated daily for eight time steps (from 00 to 21 UTC) by
the operative centre – air force meteorological service (COMET).

FARM version 4.7 (Gariazzo et al., 2007; Silibello et al., 2008; Cesaroni et al., 2013; Calori et al., 2014) is a four-dimensional Eulerian model for simulating the transport, chemical conversion and deposition of atmospheric pollutants with 1 km spatial resolution, 1-hour temporal resolution and 16 different vertical levels (from the surface to 9290 m). FARM can be easily interfaced to most available diagnostic or prognostic NWP models, as done with COSMO in the present work. Pollutants
emission from both area and point sources can be simulated by FARM, taking transformation of chemical species by gas-phase chemistry (Carter, 2000), dry and wet removal into account. Particularly interesting for our study is the aerosol module (AERO3_NEW), coupled with the gas-phase chemical model and treating primary and secondary particle dynamics and their interactions with gas-phase species, thus accounting for nucleation, condensational growth and coagulation (Binkowski, 1999). Three size modes are simulated independently: the Aitken mode (D < 0.1 μm), the accumulation mode (0.1 μm < D < 2.5 μm)
and the coarse mode (D > 2.5 μm). A national (QualeAria, here referred to as "boundary conditions") and a regional ("local sources", updated to 2015) emission inventories are supplied to the CTM to accurately assess the magnitude of the pollution loads and their variability in both time and space.

The forecasted wind velocity profile from COSMO was here used as an input parameter into the publicly-available LA-GRANTO Lagrangian analysis tool, version 2.0 (Sprenger and Wernli, 2015), to numerically integrate the high-resolution 3-D
wind fields and to determine the origin of the air masses sampled by the ALC over Aosta–Saint Christophe. We set up the program to start 8 trajectories in a circle of 1 km around the observing site and at 7 different altitudes from the ground to 4000 m a.s.l., for a total of 56 trajectories for every run. A backward run time of 48 hours was considered sufficient, on average, to cover most of the domain of the meteorological model, while keeping the numerical errors with increasing running time contained.

### 30 3.2 Trajectory Statistical Models

Back-trajectories calculated with LAGRANTO were also employed in Trajectory Statistical Models (TSMs) to provide a general picture of the geographical distribution of the probable aerosol sources. In this broadly used technique, the NWP domain is divided in cells by a grid ($i$ and $j$ indexes), and air parcels arriving to a specific receptor site are analysed. When a



cell $ij$ is crossed by a back-trajectory $l$, the tracer concentration $c_l$ measured at the arrival point of that trajectory (receptor) is considered. Finally, for each cell, a weighted average $P_{ij}$ is calculated as follows to yield a map of the possible source areas (e.g., Kabashnikov et al., 2011):

$$P_{ij} = \frac{\sum_{l=1}^{L} F(c_l)\tau_{ij}(l)}{\sum_{l=1}^{L} \tau_{ij}(l)} \qquad (2)$$

where $L$ is the total number of trajectories and $\tau_{ij}$ the time spent by the trajectory $l$ in the grid cell $ij$. $F(c_l)$ is a function of the concentration at the receptor, and, depending on the chosen technique, can be the concentration itself (Concentration Weighted Trajectories (CWT) method, e.g, Hsu et al., 2003), the logarithm of the concentration (Concentration Field (CF) method, Seibert et al., 1994) or the Heaviside step function $H(c_l - c_T)$, where $c_T$ is a concentration threshold (Potential Source Contribution Function (PSCF) method, e.g., Ashbaugh et al., 1985), generally the 75th percentile of the concentration
series. In this last case, $P_{ij}$ can be statistically interpreted as the conditional probability that concentrations above the threshold $c_T$ at the receptor site are related to the passage of the relative back-trajectory through the location $ij$ (e.g., Squizzato and Masiol, 2015). Additional iterative methods exist to reduce trailing effects and to better identify pollution hotspots (e.g., Stohl, 1996). Moreover, the obtained field $P_{ij}$ can be further weighted as a function of the number of trajectories passing through each cell, thus reducing the impact of cells with limited statistics (Zeng and Hopke, 1989).

Any series of atmospheric measurements can be used as the concentration variable $c_l$ in Eq. 2. In this work, we achieved especially meaningful results employing the scores from the PMF decomposition (Sect. 3.3), i.e. the contributions of the identified sources to the $PM_{10}$ daily concentration measured at Aosta–Downtown. The same approach, coupling PMF and TSMs, was employed in other recent studies (e.g., Bressi et al., 2014; Waked et al., 2014; Zong et al., 2018). However, since only daily average information is available from the chemical speciation, the PMF output was repeated 8 times per day in
order to allow correspondence with the 8 back-trajectories per day issued by COSMO and LAGRANTO. All methods (CWT, CF, PSCF) were explored and provided similar results. Therefore, since TSMs were used in this study to provide a general idea of the pollutant distribution and we did not aim at exhaustively discussing the differences among all existing techniques, only results of the CF method were presented. The COSMO-I2 domain was thus divided in 130x90 grid cells and 48h back-trajectories (interpolated to 5 minutes) were considered. Only points of the back-trajectories at altitudes lower than 2000 m
a.s.l. were taken into account, i.e. the approximate average height of the mixing layer in the Po basin (Diémoz et al., 2019, and Fig. 20f in the present paper). Similarly, only trajectories ending at altitudes lower than 1500 m a.s.l. over Aosta were examined, since the arriving air parcels should be able to impact on surface PM levels (the model surface altitude in the Aosta area being about 900 m a.s.l.). Weightings linearly varying from 0 (for $N < 20$ end points in a cell) to 1 (for $N > 200$) were applied to the final concentration field to reduce statistical noise.

**3.3   Positive Matrix Factorisation**

The Positive Matrix Factorisation (PMF,  Paatero and Tapper, 1994; Paatero, 1997) technique, as implemented in the US EPA PMF5.0 software (Norris and Duvall, 2014), was employed to study the 2017 series of aerosol chemical analyses and air





quality measurements in Aosta–Downtown. The method allowed us to identify the possible emission sources of the observed aerosol, and notably to discriminate its local and non-local origin. PMF requires a long multivariate series of chemical analyses, matrix $\mathbf{X}$ (the $n$ rows being the samples and $m$ columns representing the chemical species) and decomposes it into the product of two positive-definite matrices, i.e. $\mathbf{G}$ ($n \times p$, matrix of factor contributions, $p$ being the number of factors chosen for the decomposition) and $\mathbf{F}$ ($p \times m$, matrix of factor profiles) such that

$$\mathbf{X} = \mathbf{G}\,\mathbf{F} + \mathbf{E} \qquad\qquad (3)$$

A solution is found by minimizing the squared sum of the residuals (matrix $\mathbf{E}$), weighted by their respective measurement uncertainties (this objective function is called $Q$). A total of 100 runs were performed in this work for each dataset to find an optimal solution. Here we considered three different series $\mathbf{X}$: a) the overall dataset of anion/cation analyses (data available almost every day in 2017, $n = 360$, "PMF-dataset a"), b) a subset of dataset a), selecting those measurements having coincident EC/OC analyses ($n = 132$, "PMF-dataset b") and c) a subset of a), selecting those measurements having coincident metal speciation ($n = 209$, "PMF-dataset c"). To make the decomposition of the three series comparable, an "unidentified" contribution, corresponding to the carbonaceous species only included in PMF-dataset b, was added to PMF-datasets a and c. This was done in the following way: we first estimated the total mass of crustal elements using the measured water-soluble calcium concentration as a proxy, with an empirical conversion factor of 10 (e.g., Waked et al., 2014, a more rigorous estimate for the soil component, confirming this factor, was obtained from the PMF analysis itself); we then subtracted the sum of the available chemical components (including the estimated mass from soil components) from the total measured $PM_{10}$ concentration, thus closing the mass balance. This difference was then added in PMF-datasets a and c as an unknown source.

NO, $NO_2$ and total PAHs measured at the same site (Aosta–Downtown) were furthermore included in the PMF to help identification of local pollution sources. The number of factors for each dataset was chosen based on the $Q/Q_{exp}$ ratio and physical interpretability of the resulting factors (Sect. S4). The measured $PM_{10}$ concentration was selected as the total variable to determine the contribution of each mode to the $PM_{10}$ concentration.

PMF was additionally performed on volume size distributions measured by the Fidas OPC in Aosta–Saint Christophe (Sect. 4.3.3), the $m$ columns representing sizes, instead of chemical species.

## 4 Results

### 4.1 Daily classification schemes based on ALC measurements or meteorology

As the vertical dimension was a key factor in identifying the events of aerosol pollution transport over the northwestern Alps (e.g., Diémoz et al., 2019), a first step of our analysis was to set up a classification scheme of those advection dates based on ALC measurements. To this purpose, we used the longest ALC record available (2015–2017) and coupled it to meteorological measurements/forecasts. Since, to our knowledge, no previous ALC-based classification method is available in the scientific literature, we defined an original, daily-resolved classification scheme based on ceilometer measurements to



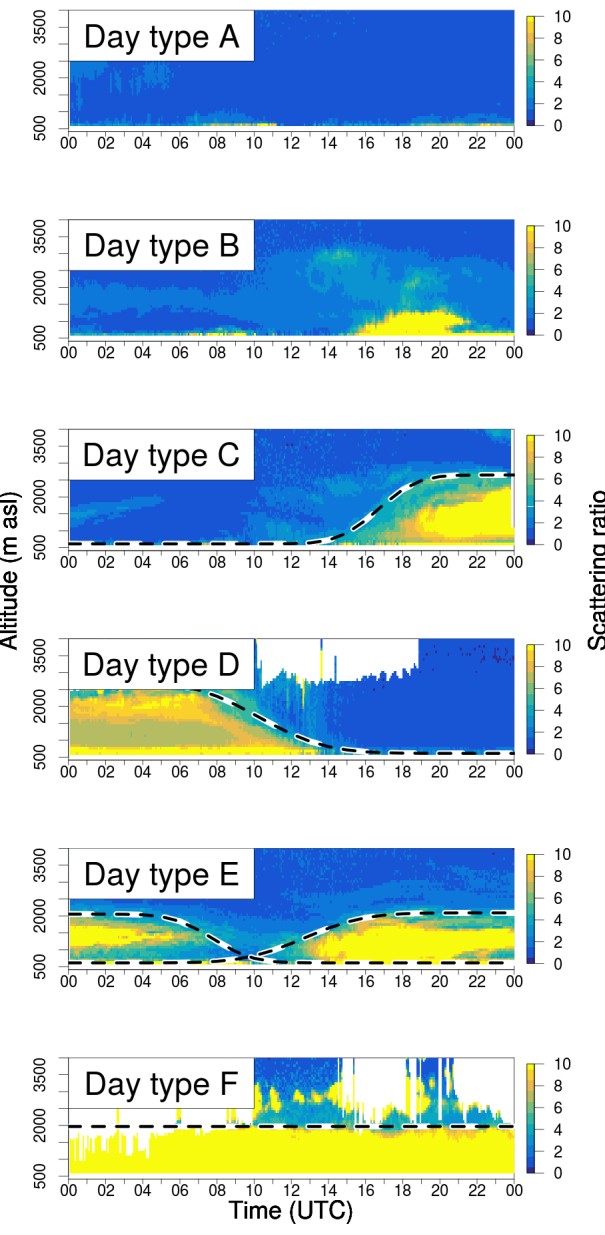

**Figure 2.** Example of ALC images representative of each category (A-F) described in Sect. 4.1. Corresponding dates are 1 December 2015, 19 October 2016, 20 April 2016, 11 April 2015, 1 November 2017 and 27 January 2017. The dashed lines for categories C–F identify the sigmoid interpolation to the SR=3 envelope using the automated algorithm as explained in Sect. 4.2.





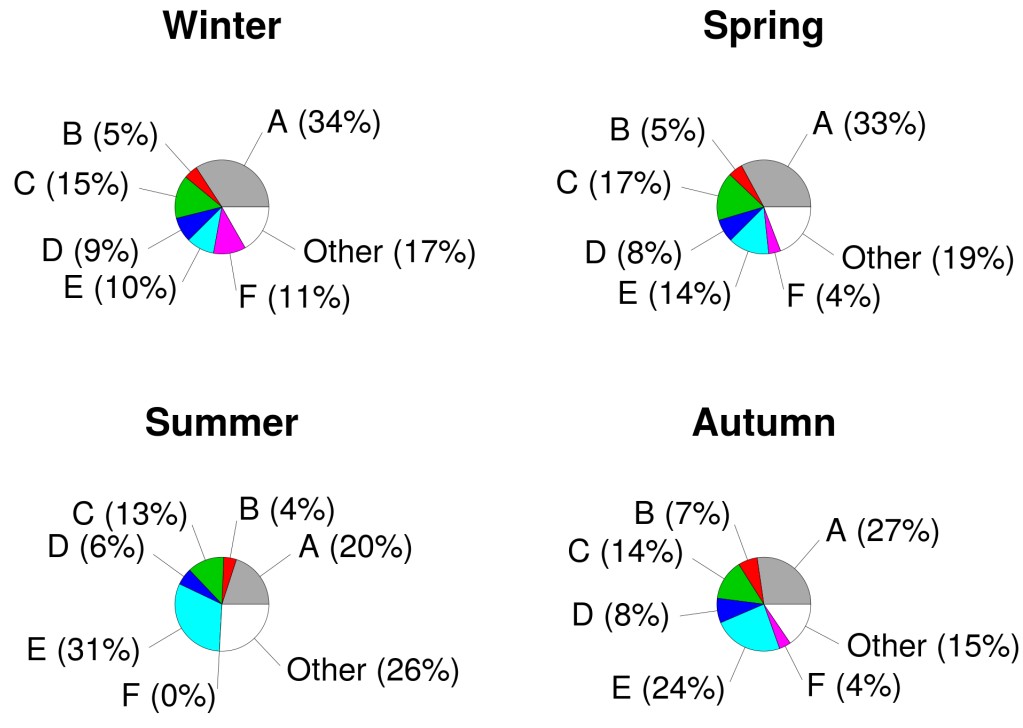

**Figure 3.** Frequency distribution of the aerosol layers observed in Aosta–Saint Christophe based on the ALC classification described in Sect. 4.1. A=no layer, B=short episodes, C=layer detected in the afternoon, D=leaving layer, E=layer dissolving in the morning and a separated structure in the afternoon, F=persistent layer.

group dates according to specific ALC-observed conditions. A graphical overview with examples for each class is given in Fig. 2. Overall, we identified six types (classes A–F) of days:

– days "A": no aerosol layer or only low (i.e., $< 500$ m from the ground) aerosol layers visible for the whole day. These days are intended to represent unpolluted conditions or cases only affected by local sources, since aerosol transport from remote regions generally manifests as more elevated layers, as found by Diémoz et al. (2019);

– days "B": only brief episodes of layers developing from the surface to altitudes $> 500$ m a.g.l. observed during the 24 hours. The origin of the aerosol layers of this intermediate class is uncertain, since they could either result from weak (not completely developed) advections from outside the investigated region or from puffs of locally-produced aerosol. For this reason, data in class "B" were used only as a transitional level, but not to draw definitive conclusions;



**Table 2.** Classification of weather regimes used in the study, based on wind speed ($|\boldsymbol{v}|$), wind provenance, daily duration of the phenomenon ($\Delta t$) and other measured meteorological variables.

| Primary classification | Secondary (detailed) classification | Definition |
|---|---|---|
| Easterly winds | Upwind | $|\boldsymbol{v}| > 1\,\mathrm{m\,s^{-1}}$, $\Delta t > 12\,\mathrm{h}$ |
| | Breeze | Night (19–8 UTC): calm wind ($|\boldsymbol{v}| < 1\,\mathrm{m\,s^{-1}}$) or low westerly wind ($|\boldsymbol{v}| < 1.5\,\mathrm{m\,s^{-1}}$) |
| | | Day (8–19 UTC): easterly wind ($|\boldsymbol{v}| > 1.5\,\mathrm{m\,s^{-1}}$), $\Delta t > 4\,\mathrm{h}$ |
| Westerly winds | Downwind | $|\boldsymbol{v}| > 1\,\mathrm{m\,s^{-1}}$, $\Delta t > 12\,\mathrm{h}$ |
| | Foehn | $|\boldsymbol{v}| > 4\,\mathrm{m\,s^{-1}}$, $\Delta t > 4\,\mathrm{h}$, wind provenance $< 25°$ or $> 225°$, RH$< 40\%$ |
| Stability | – | $|\boldsymbol{v}| < 1\,\mathrm{m\,s^{-1}}$, $\Delta t > 20\,\mathrm{h}$ |
| Other | Precipitation | Accumulated daily precipitation $> 1\,\mathrm{mm}$, $\Delta t > 4\,\mathrm{h}$ |
| | Unclassified | Every day that does not fall in the previous categories |

- days "C": detection of an elevated, well-developed aerosol layer (usually, in the afternoon) and persistent during the night;

- days "D": detection of an aerosol layer from the previous day dissolving during the morning hours. No layers in the afternoon;

- days "E": detection of a first aerosol layer dissolving (or strongly reducing) in the morning and a separated structure in the afternoon;

- days "F": thick layer persisting all day long.

Note that, although we also developed automatic recognition algorithms for classification purposes, we finally chose a classification based on visual inspection (for a total of 928 daily images) to ensure the maximum quality of the flagging and

10 avoid further sources of error.

We used the ALC classification above to derive the seasonal frequency distribution of the aerosol advections classes. Relevant results are shown in Fig. 3. Days not falling in those classes (e.g. complex-shaped layers, presence of low clouds or heavy rain hiding the aerosol layer, etc.) or including other kind of aerosol layers (as Saharan dust) were flagged as "Other" and not considered for further analyses (this accounting to maximum 26% of dates, in summer). The results reveal that clearly-detected

15 aerosol advections (cases C to F) occur half (50%) of the days in summer and autumn and with a slightly lower frequency (45%) in winter and spring. A higher percentage of clear days (A) occurs in winter and spring, while persistent layers (F) are mostly found in winter. Conversely, cases E are more frequent in summer.

To assess how the ALC classification described above is related to the circulation patterns, we used the surface meteorological observations in Aosta–Saint Christophe as drivers of a second, independent classification scheme as summarised in Table



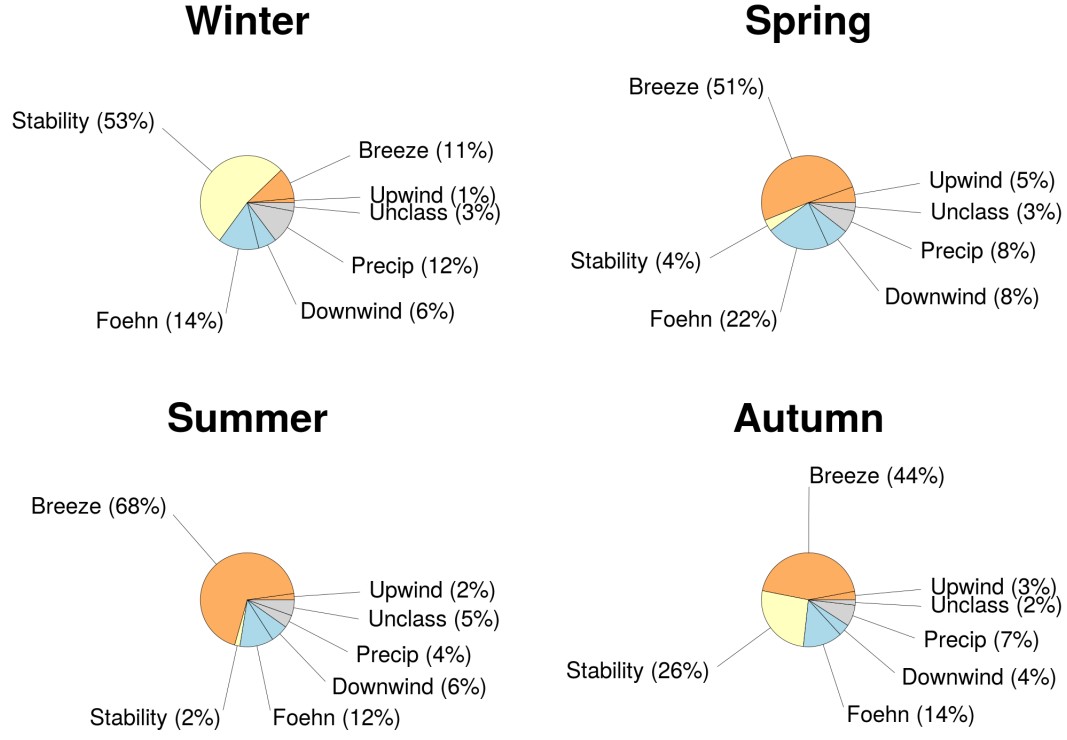

**Figure 4.** Frequency distribution of circulation conditions in Aosta–Saint Christophe based on the meteorological classification described in Table 2. Easterly winds (Breeze and Upwind) are represented as orange slices, stability conditions in yellow and westerly winds (Foehn and Downwind) in blue. Precipitation and unclassified days were not used in the study and are represented as grey slices.

2. Seasonal frequency distributions from this meteo-based scheme are shown in Fig. 4. These show that weather circulation types are remarkably variable during the year and help in interpreting the ALC-based results (Fig. 3). Winter is mostly charac-terised by stability (53% of the days), which sometimes favours stagnation and persistent aerosol layers (F). Plain-to-mountain breeze gradually increases from winter (only 11%, which reflects the high percentage of clear days A in that season) to summer

5 (68%), when the thermally-driven regional circulation represents the main mechanism contributing to transport of polluted air masses (cases E). Foehn (rain shadow) winds are fundamental processes, especially in spring (22%), leading to the removal of pollutants and frequent occurrence of clear days (A) in that season. Finally, upwind or downwind synoptic fluxes are also partly responsible for pushing polluted air masses respectively towards or away from the Aosta Valley, although they are markedly less frequent compared to the breeze mechanism.

10  The association between the ALC- and meteo-based classification schemes was explored using contingency tables (Fig. 5a). To this aim, the ALC classification (A–F) was further simplified and reduced to two main cases: presence of an arriving or




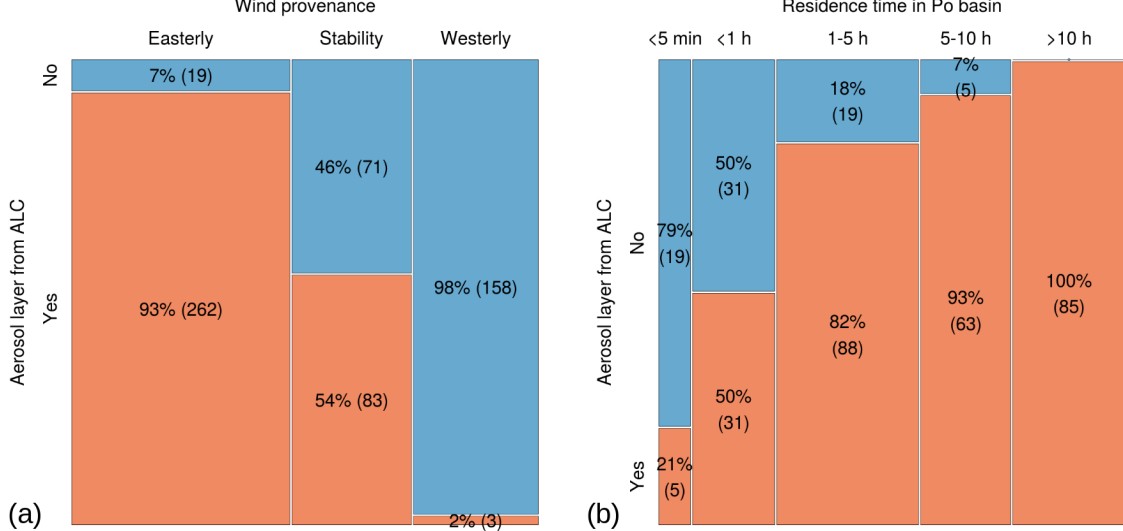

**Figure 5.** (a) Contingency table showing occurrences of the aerosol layer seen by the ALC and wind provenance. The blue boxes represent cases when the aerosol layer is not visible (day types A) and pink boxes cases when an arriving or stationary aerosol layer is revealed by the ALC (day types C, E and F). The numbers inside the boxes refers to the frequency and the number of occurrences for each couple of wind/layer classes. (b) Occurrence of the aerosol layer seen by the ALC and residence time of 48h back-trajectories in the Po plain.

stationary aerosol layer (classes C, E and F) and absence of any elevated aerosol layer (class A). Classes B (short and uncertain episodes) and D (layer leaving during the morning) were not used for the next considerations to avoid confunding conditions. Figure 5a shows that the presence of an elevated aerosol layer is strongly connected to the wind provenance, as expected. In fact, this scheme reveals that the layer appearance can be easily explained using the wind direction as the only information: when

air masses come from the east, the probability of identifying an aerosol layer over the Aosta Valley is of 93%, confirming the Po basin as an heavy aerosol source for the neighbouring regions, irrespectively of the day and period of the year. Conversely, this probability reduces to only 2% of the cases when the wind blows from the west. In the period addressed, we thus detected few exceptions to the perfect correspondence (100% and 0%), which can however be explained. For cases of easterly winds and no aerosol layer detected, possible reasons are:

– the up-valley breeze, although clearly detected by the surface network, was too short or too weak to transport the polluted air masses from the Po basin to Aosta–Saint Christophe (at least ∼40 km must be travelled by the air masses along the main valley). This was for example the case of 14 and 18 September 2015, and 17 June 2016;

– back-trajectories were indeed channelled along the central valley, however they did not come from the Po basin, but rather from the pre-Alps. This was the case, for instance, of 20 August 2015, in which air masses came from the Divedro

Valley.





There are few cases in our record in which, conversely, an aerosol layer was associated to westerly, rather than easterly, winds. These are as follows:

– on 5 February 2016 and 15 March 2016, the prevalent wind was from the west and the days were correctly identified as '"Downwind" and "Foehn", respectively. However, as soon as the wind turned and the valley-mountain breeze started,
the aerosol layer arrived;

– on 24 March 2016, a real case of transboundary/transalpine advection occurred and an aerosol-rich air mass was transported from the polluted French valleys on the other side of the Mont Blanc chain to the Aosta Valley. Although heavy pollution episodes can occur also on the French side of the Alps (e.g., Chazette et al., 2005; Brulfert et al., 2006; Bonvalot et al., 2016; Chemel et al., 2016; Largeron and Staquet, 2016; Sabatier et al., 2018), it is very rare to clearly spot an
aerosol layer transported from that region to Aosta, since air masses coming from the west must have crossed the Alps and are almost always mixed with clean air from the uppermost layers, therefore only a dim aerosol backscatter signal is usually noticed from the ALC.

Overall, the good correspondence between the measured wind regimes and the aerosol layer detection by the ALC (Fig. 5a) suggests that the same association can be employed to forecast the arrival of an advected aerosol layer based on the predicted
wind fields. As an example of simple, forecasting capabilities of this phenomenon, Fig. 5b illustrates a contingency table relating the occurrence of an aerosol layer in Aosta–Saint Christophe to the residence time over the Po Valley of the 48-hours back-trajectories arriving to the Aosta–Saint Christophe observing site. Again, a direct, clear connection between both variables can be seen, with a 100% correspondence for residence times of air masses over the Po basin >10 hours.

Finally, it is worth mentioning that the proven high correlation between the circulation patterns (observed and/or simulated)
and the presence of advected polluted layers in the Aosta area may be exploited in the future to reconstruct the impacts of aerosol transport on air quality over the Alpine region back to previous periods not covered by the ALC measurements (a 40-year-long meteorological database is available in the region).

## 4.2    Vertical and temporal characteristics of the advected aerosol layer

The 2015–2017 ALC time series in Aosta–Saint Christophe is summarised in Fig. 6, showing monthly averages of the SR daily
evolution. To draw the figure, the data were first grouped in 1-hour bins (clouds were removed as well as periods with $< 30$ minutes of measurements per hour), then monthly means were calculated (excluding points with $< 1$ week of measurements per month). The plot clearly shows that the maximum aerosol backscatter is generally found in the early morning and in the late afternoon, likely due to coupling between aerosol transport and hygroscopic effects (Diémoz et al., 2019), and not, as usual for plain sites, in correspondence of the midday development of the mixing boundary layer. Also, the altitude of the
layer varies with the season. Incidentally, months affected by exceptional events can be sharply distinguished, and namely the frequent Saharan dust events in June and August 2017, and the forest fire plumes from Piedmont in October 2017, this latter again affecting the Aosta site in the afternoon, because of the breeze circulation. A similar climatology, showing the results







**Figure 6.** Monthly averages of the SR daily evolution (hourly measurements, from 0 to 24 UTC) from the ALC in Aosta–Saint Christophe (2015–2017). Although ALC measurements extend up to 15 km altitude, we limited the figure to 5000 m a.g.l. to better show the lowermost levels, where aerosol transport from the Po basin occurs (Diémoz et al., 2019). Points with insufficient statistics are not plotted (white areas, cf. main text).



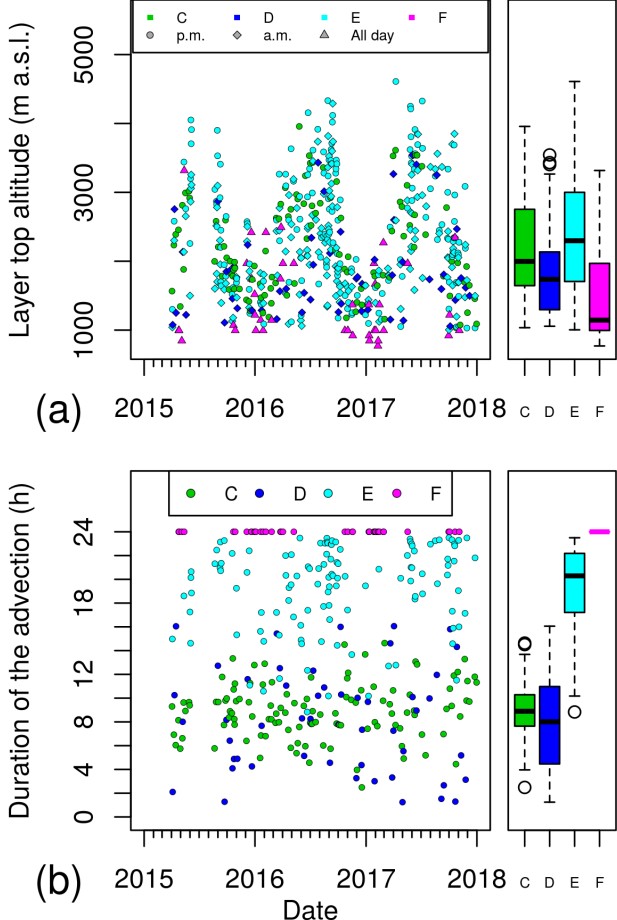

**Figure 7.** (a) Aerosol layer top altitude for the classified days. Marker colour represents the type of the advection, while the marker shape represents the time of the day: morning (a.m.), afternoon (p.m.) or all day (applicable to cases F only). (b) Overall duration of the aerosol layer in a day (morning and afternoon durations were summed up in case E). The boxplots at the right represent synthetic information for each day type (same y-scale as the relative charts at the left). Missing measurements in summer 2015 are due to the replacement of the ALC laser module.

in terms of average PM concentration profiles retrieved by the ALC is given in Fig. S1 of the Supplement. In that case, the conversion from backscatter to $PM_{10}$ was obtained using the methodology described by Dionisi et al. (2018).

To quantitatively explore the spatial and temporal characteristics of the ALC-detected aerosol layer over the long term, we developed an automated procedure, described in detail in the Supplement (Sect. S2), to identify and fit with a sigmoid curve the space-time region of the ALC profiles affected by the aerosol advection (using SR≥ 3 as threshold value). This allowed to objectively determine, for example, the height and the duration of the advections. The long-term statistics of these two variables is summarised in Fig. 7. The altitude of the polluted aerosol layer (Fig. 7a) displays, as expected, a clear seasonal



cycle, with minimum in winter and maximum, up to 4000 m a.g.l., in summer. The overall cycle mimics the variability of the convective boundary layer (CBL) height found in the Po basin by other studies (Barnaba et al., 2010; Decesari et al., 2014; Arvani et al., 2016), with somewhat higher values that reflect the modification of the boundary layer structure in mountainous areas (Teixeira et al., 2016). Notably, stronger, multi-scale thermally-driven flows (e.g. the ones developing on the slopes of

the valley) further redistribute the aerosol particles in the vertical direction, thus pushing the upper limit of the CBL to the ridge height. Average values of the different classes, represented as box plots in Fig. 7, are clearly impacted by the season of their maximum frequency over the year (Fig. 3). In fact, minimum altitude of the layer top was found in days F owing to their maximum occurrence in winter, while top altitude maximum was found in days E being these mostly summertime events. Great variability was also found for the duration of the advection (Fig. 7b), ranging from few hours in the weakest events to

24 hours/day in the extreme cases (full day, by definition, for cases F). The duration of the events, together with their absolute aerosol load, are expected to be the major drivers of the impact of the investigated phenomenon on surface air quality. This is investigated in the following paragraphs.

### 4.3 Impact of Po Valley advections on northwestern Alps surface-level PM loads and physico-chemical characteristics

To quantify the impact of the aerosol layers detected by the ALC on the air quality at ground level, we first investigated

how the PM concentration daily evolution measured by the surface network changes depending on the ALC-identified classes (Sect. 4.3.1). We then used the Positive Matrix Factorisation of the multivariate dataset of chemical analyses to explore the chemical markers associated to the transport from the Po basin (Sect. 4.3.2). The effectiveness of the identified markers and the coupling between chemistry and meteorology were also confirmed by the use of Trajectory Statistical Models. Finally, a link between aerosol chemical and physical properties was investigated in Sect. 4.3.3, where particle chemical composition is cou-

pled to measurements of aerosol particle size. Finally, a preliminary assessment on the effect of the investigated advections on surface pollutants other than particulate matter (e.g., $NO_x$ partitioning) was also performed and is reported in the Supplement (Sect. S5).

### 4.3.1 Surface PM variability in relation to the ALC profiles classification scheme

We started investigating the effect of aerosol transport on the Aosta Valley surface air quality by analysing the variations of the

daily mean PM concentrations measured by the ARPA surface network as a function of the ALC classes introduced in Sect. 4.1. Figure 8a shows the relevant results, resolved by season. It is quite evident that $PM_{10}$ concentrations in Aosta–Downtown are in fact highly correlated with the presence/strength of the aerosol layers seen by the ALC. $PM_{10}$ on days with a persistent aerosol layer (class F) was found to be 2 to 3.5 times higher, on average, than on non-event days (class A, this difference being statistically significant for all seasons, p-value$< 0.05$ from the Mann-Whitney test (Mann and Whitney, 1947)). Similar

behaviour was seen in $PM_{2.5}$ concentrations measured at the same station (Fig. S2a) and in the $PM_{2.5}/PM_{10}$ ratio (Fig. S2b), this latter indicating that aerosol particles are smaller, on average, on days when the ALC detects a thick aerosol layer compared to non-event days. Causality between the presence of an elevated layer and variations of the PM surface concentration, however, cannot be rigorously inferred at this point, since, in principle, common environmental conditions affecting both PM at ground





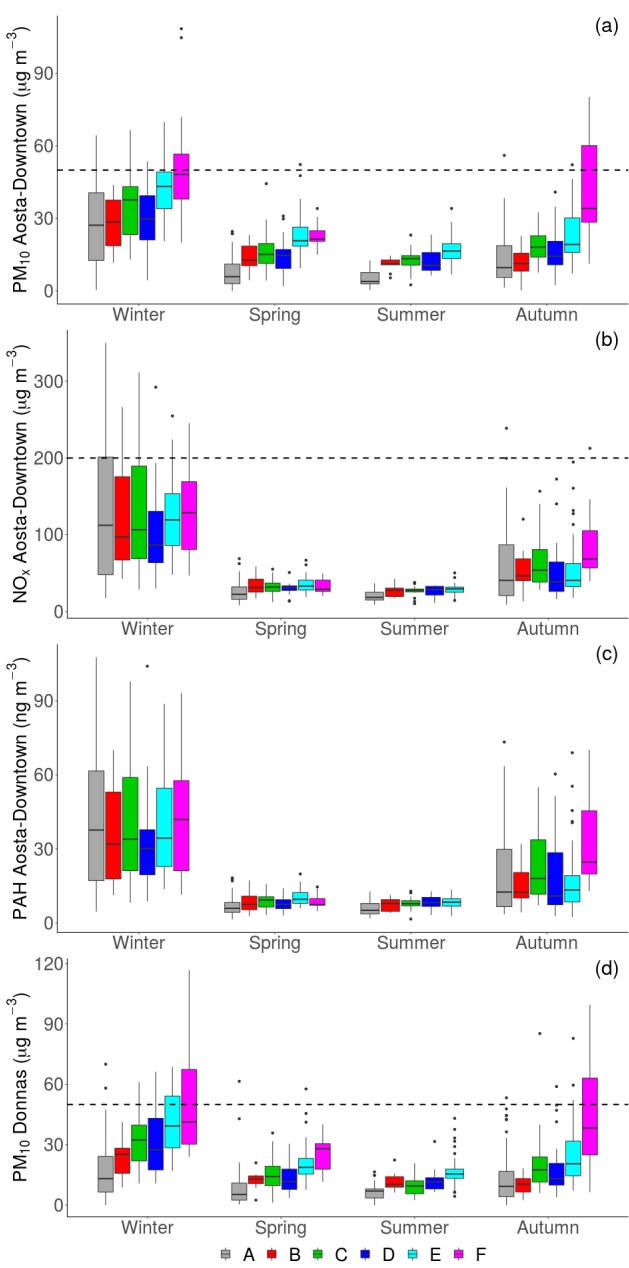

**Figure 8.** Daily-averaged surface concentrations (2015–2017) of $PM_{10}$ (a), nitrogen oxides (b) and polycyclic aromatic hydrocarbons (c) in Aosta–Downtown as a function of the ALC classes and season. (d) $PM_{10}$ surface concentration at the Donnas station. The EU-established $PM_{10}$ daily limit of 50 $\mu g\,m^{-3}$ and the $NO_2$ hourly limit of 200 $\mu g\,m^{-3}$ are also drawn as references (horizontal dashed lines).



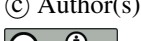

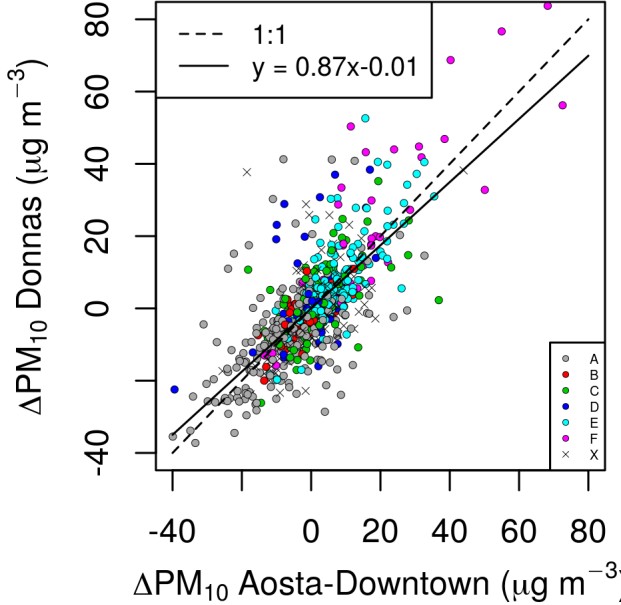

**Figure 9.** Comparison between daily $PM_{10}$ anomalies (Eq. 4) registered in Aosta–Downtown and Donnas (40 km apart), relative to a monthly moving average. Circles colour represents the ALC classification, see legend, the "X" symbol represents the unclassified days. The 1:1 and the best fit line are also represented on the plot. The Pearson's correlation index between the two series is $\rho = 0.73$.

and the aerosol in the layers aloft could originate the observed correlation. Nevertheless, it is interesting to note that this effect is less detectable for other pollutants measured by the network (Figs. 8b and c). In fact, concentrations of typical locally-produced pollutants such as $NO_x$ and PAHs do not change as much as PM among the ALC classes, with only rare exceptions (e.g., class A, which is slightly influenced by foehn winds, and class F, in which temperature inversions/low mixing layer heights may

5  enhance the effect of local surface emissions). These results point towards excluding that the correlation trivially originates from common weather conditions or uneven distribution of the ALC classes among the seasons.

Furthermore, large correlation was also found between the ALC classification obtained in Aosta–Saint Christophe and the surface $PM_{10}$ concentration recorded in Donnas (Fig. 8d). Since the two stations are located at 40 km distance, this result suggests that a major role is played by large-scale dynamics rather than by local sources and that the phenomena under

10  consideration have at least a regional-scale extent. As a further test, we correlated the daily $PM_{10}$ concentrations measured in Donnas to those measured in Aosta. To this purpose, in order to avoid large, fictitious correlations due to the same yearly cycle (maximum PM concentration in winter and minimum in summer), and only focus on the short-term dynamics, we considered the anomaly ($\Delta PM_{10}$) with respect to the monthly moving average ($\langle PM_{10}\rangle_{\mathrm{m}}$), i.e.:

$$\Delta PM_{10}(t) = PM_{10}(t) - \langle PM_{10}\rangle_{\mathrm{m}}(t) \tag{4}$$




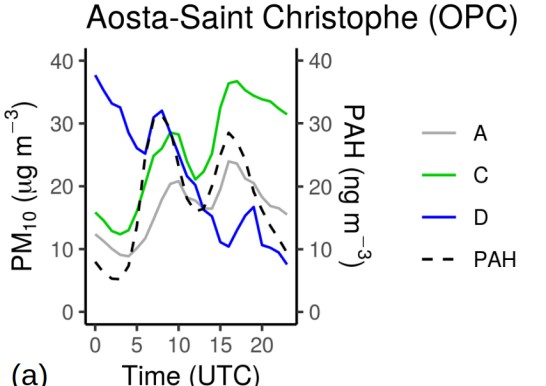
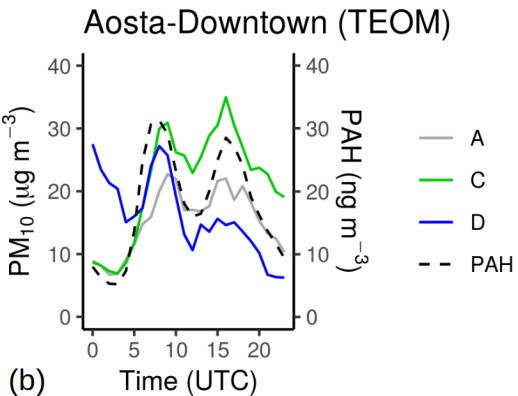

**Figure 10.** (a) Daily $PM_{10}$ cycle sampled by the OPC in Aosta–Saint Christophe during non-event days (class A) and days with arriving and leaving aerosol layers (classes C and D), plotted together with the daily evolution of PAH as a proxy of the local sources (dotted line, right vertical axis, in $ng\,m^{-3}$). (b) Same as (a) using the TEOM in Aosta–Downtown.

where $t$ is a specific day. The scatterplot of these anomalies is shown in Fig. 9. It highlights the good correlation between the $\Delta PM_{10}$ at the two sites ($\rho = 0.73$ when considering all classes and $\rho = 0.76$ if only advection cases, i.e. classes C to F, are considered). Using the ALC classification to cluster these results (circle colours) further helps showing: 1) the different $\Delta PM_{10}$ associated to the different classes and 2) the good correspondence of this effect at both sites. Notably, in the most severe cases

(E and F), the anomalies in Donnas are generally larger than in Aosta–Downtown (the best fit line being $y = 1.1x + 0.8$ for cases E and F only), which is compatible with this site being closer to the Po basin (Fig. 1).

The advected aerosol layers were also found to impact the daily evolution of surface PM concentrations. To investigate this aspect, we used hourly and sub-hourly $PM_{10}$ data from both the Fidas 200s OPC in Aosta–Saint Christophe and the TEOM in Aosta–Downtown. The $PM_{10}$ daily cycles are presented for cases A (no advection), C and D only, for sake of clarity,

and for both Aosta–Saint Christophe (Fig. 10a) and Aosta–Downtown (10b). Despite the instruments limitations described in Sect. 2, the figure shows that local effects play an important role on the $PM_{10}$ daily evolution, as visible from the morning and afternoon peaks. In fact, these maxima, common to most pollutants, are the results of the coupled effect of varying local emissions and boundary layer height during the day. The daily cycle of PAH concentrations is added to the plot as a proxy of diurnal cycle from local sources (dotted line, right y-axis). This cycle is very similar to the $PM_{10}$ one for case A. Conversely,

the influence of the advections (C and D) is clearly visible on the corresponding $PM_{10}$ cycles. In fact, Fig. 10 shows the $PM_{10}$ concentration to markedly increase in the afternoon of days C (by 1.0 and 0.7 $\mu g\,m^{-3}\,h^{-1}$ in Saint-Christophe and Aosta–Downtown, respectively) and to markedly decrease in days D (by -1.3 and -0.7 $\mu g\,m^{-3}\,h^{-1}$). This effect is similar at the two sites although less marked in Aosta–Downtown, especially at night. This is probably due to TEOM loss of volatile species in the advected aerosol (see also the PMF results on chemical analysis in the following section). Similar results (not shown) were

obtained when splitting the data on a seasonal basis, which excludes that the observed behaviour originates from the seasonal cycle.



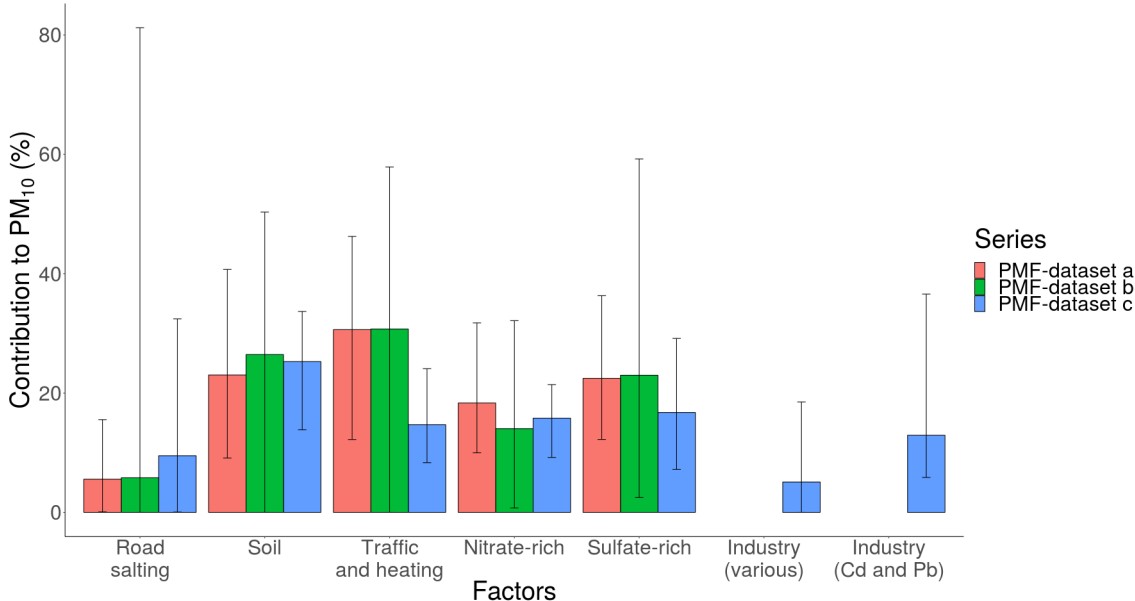

**Figure 11.** Percentage of aerosol mass concentration carried by each mode for the three PMF-datasets considered in the analysis (PMF-dataset a: anion/cation only; PMF-dataset b: anion/cation together with EC/OC; PMF-dataset c: anion/cation together with metals), and their respective confidence intervals from the BS-DISP test. Details are provided in Sect. S4.

### 4.3.2 Chemical species within the advected aerosol layers

As a further step to adequately discriminate local and non-local sources, especially when dealing with data collected in an urban environment such as Aosta–Downtown, we took advantage of the one-year-long (2017) chemical aerosol characterisation dataset in Aosta–Downtown and applied the PMF to the three datasets described in Sect. 3.3 (PMF-datasets a, b, c, i.e.,

anion/cation only, anion/cation with EC/OC and anion/cation with metals, respectively). The main factors shaping the composition of $PM_{10}$ at the investigated site are given in Fig.11, all details on this outcome being given in Sect. S4. The question addressed here is how aerosol transport from the Po valley affects the chemical properties of $PM_{10}$ sampled in Aosta. In the preliminary analysis of three cases studies reported in the companion paper, we found that the advected layers were rich of nitrates and sulfates (accounting, together with ammonium, for 30-40% of the total $PM_{10}$ mass). This kind of secondary inor-

ganic aerosol was indeed found in high concentrations in the Po basin by previous studies (e.g., Putaud et al., 2002; Carbone et al., 2010; Putaud et al., 2010; Larsen et al., 2012; Saarikoski et al., 2012; Gilardoni et al., 2014; Curci et al., 2015). Here we further tested on a longer dataset if the sum of the nitrate- and sulfate-rich factors (i.e. secondary aerosols) could in fact represent a good chemical marker of aerosol transport from the Po basin (e.g., Kukkonen et al., 2008; Tang et al., 2014). In this respect, it is worth highlighting that these PMF modes are not correlated with typical locally-produced pollutants such

as $NO_x$ and PAHs (this is revealed by the low percentage contribution of $NO_x$ and PAHs in nitrate-rich and sulfate-rich





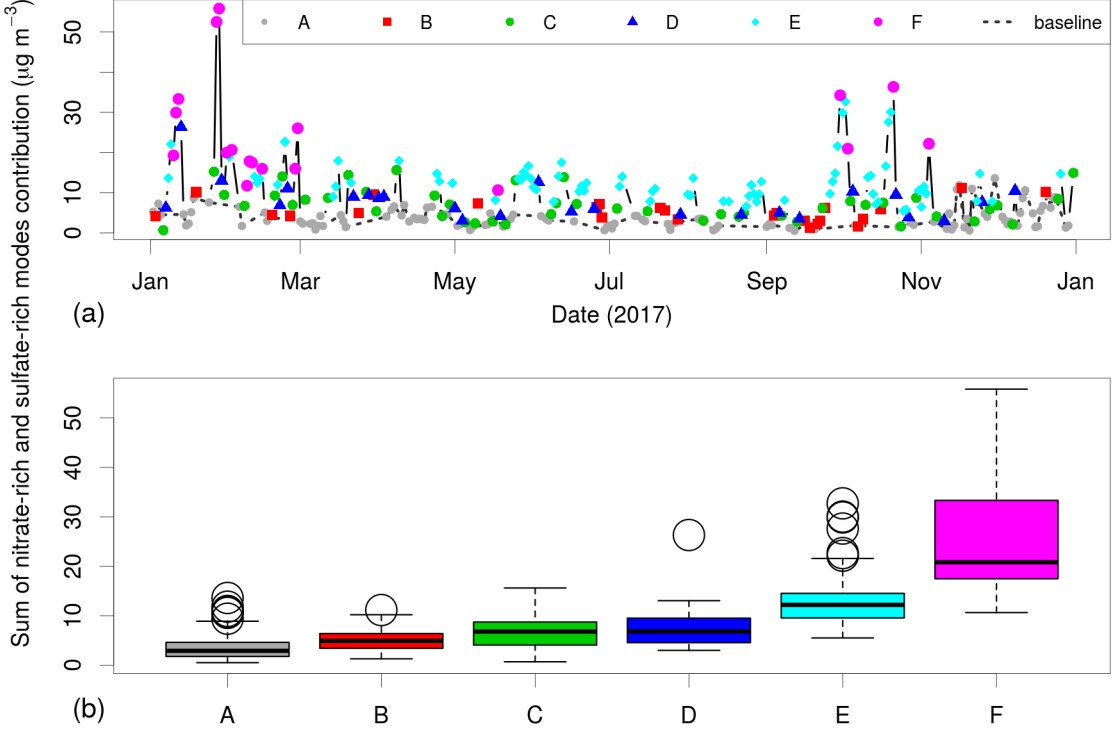

**Figure 12.** (a) Temporal chart of the contribution of secondary (nitrate-rich and sulfate-rich) modes to the total $PM_{10}$. The dashed line represents the estimated local production of secondary aerosol. The difference between each dot and the baseline can thus be read as the non-local contribution to the aerosol concentration in Aosta–Downtown. Since ions chemical speciation started in 2017, only one year of overlap with the ALC is available at the moment (see Table 1). (b) Boxplot of the contribution of secondary (nitrate-rich and sulfate-rich) modes to the total $PM_{10}$ concentration as a function of the day type. PMF-dataset "a" was used for both plots.

modes in Figs. S3–S5), which therefore excludes their local origin. We then combined the chemical information (the sum of the secondary components) with the ALC-classification. This was done for year 2017, which is the only overlapping period between anion/cation analyses and ALC profiles (see Table 1). Results are shown in Fig. 12. In particular, Fig. 12a shows the time evolution of the mass concentration carried by the secondary aerosol modes, in which each date is coloured according

5    to the six ALC classes identified. It can be noticed that almost all peak values occur during days of type E or F, i.e. when the advected layer is more persistent and able to strongly influence the air quality at ground. The very good correspondence between the sum of the nitrate- and sulfate-rich modes contributions and the ALC classification, as well as the poor correlation between the latter and the role of the other PMF-identified modes (not shown), suggests that the link between the secondary components and the aerosol layer advection is not trivially due to particular weather conditions affecting both variables. Rather,

10    the coupling between the chemical and the ALC information indicates that aerosols of secondary origin dominate during the advection episodes from the Po basin.





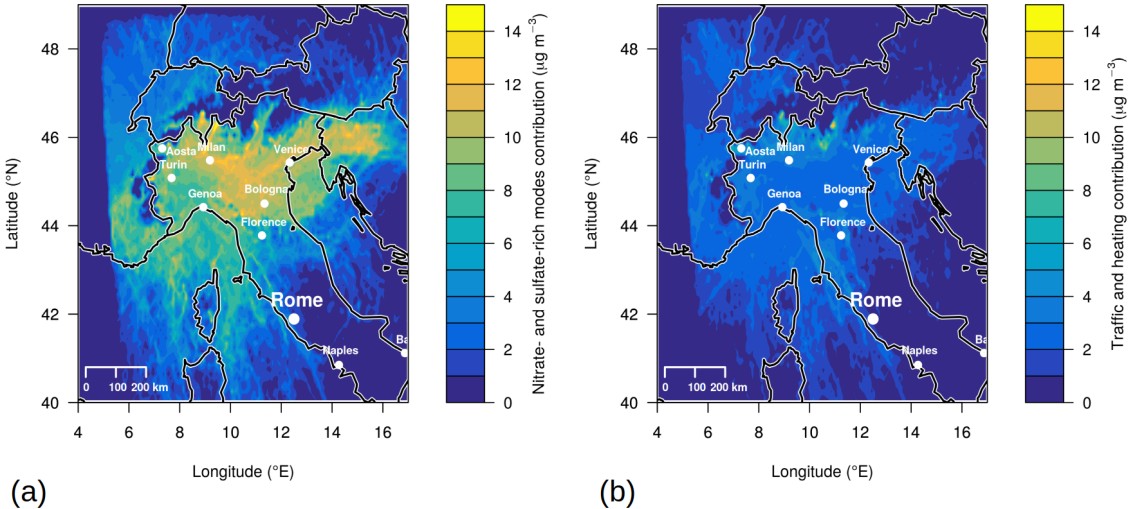

**Figure 13.** (a) Output of the Concentration Field TSM, using the sum of the contributions from PMF nitrate- and sulfate-rich modes as concentration variable at the receptor (Aosta–Downtown). (b) CF based on the traffic and heating mode from PMF. Trajectories are cut at the borders of the COSMO-I2 domain.

A summary of the contribution of secondary modes to the total $PM_{10}$ as a function of the day type, on a statistical basis, is given in Fig. 12b. The Mann-Whitney test, applied on these data, confirms that the contribution of secondary pollutants is significantly lower for class A compared to B (p=0.0005, $3 \cdot 10^{-8}$), C compared to D (p=$6 \cdot 10^{-8}$), D compared to E (p=$9 \cdot 10^{-7}$), and E compared to F (p=$6 \cdot 10^{-7}$). Figure 12b also allows to quantify the contribution of non-local sources using as baseline the

results obtained for class A (i.e., no advection, dashed line in Fig. 12a). Note that this baseline is very low on average, about 3 $\mu g \, m^{-3}$, as expected from the unfavourable conditions for secondary aerosol production in the area under investigation (e.g., low expected emissions of ammonia, frequent winds, unobserved correlation between secondary aerosol components and their gas-phase precursors). The non-local contribution is calculated as the average difference between the mass from the secondary modes and this baseline, and amounts to about 5 $\mu g \, m^{-3}$, i.e. 24% of the total $PM_{10}$ concentration from the PMF. Moreover,

advections from the Po basin account for an increase of $PM_{10}$ concentrations up to 50 $\mu g \, m^{-3}$ in specific episodes. The most important compounds contributing to the increase are nitrate, ammonium and and sulfate, i.e. the species characterising the nitrate- and sulfate-rich modes. However, since the sulfate-rich mode additionally includes organic matter (OM, cf. Figs.S3– S5), the latter species is also relevant in the advection episodes, especially in summer, when the nitrate-rich mode has a lower contribution. This finding agrees with previous works, identifying the Po basin as a source of organic aerosol (Putaud et al.,

2002; Matta et al., 2003; Gilardoni et al., 2011; Perrone et al., 2012; Saarikoski et al., 2012; Sandrini et al., 2014; Bressi et al., 2016; Khan et al., 2016; Costabile et al., 2017).

The Po Valley origin of the secondary components was also further proved coupling the chemical information to back-trajectories. Figure 13a shows the result of the TSM using the sum of the nitrate- and sulfate-rich modes as concentration variable at the receptor. It clearly reveals that trajectories crossing the Po basin have a much higher impact on secondary





aerosol measured in Aosta compared to air parcels coming from the northern side of the Alps. Interestingly, this is not the case using different source factors from PMF. For example, Fig. 13b reports the same map, but relative to the traffic and heating mode, which is clearly much more homogeneous compared to Fig. 13a and essentially reflects the density distribution of the trajectories. This behaviour highlights, as expected, the local origin of the PMF source factors other than the two (secondary
aerosols) chosen as proxies of the advection.

### 4.3.3   PMF of aerosol size distributions and links to chemical properties

Exploring the links between aerosol chemical and physical properties is useful to get insights into the atmospheric mechanisms leading to particles formation and transformation during their transport to the Aosta Valley. Therefore, we investigated correlations between the results of the chemical analyses on samples collected in Aosta–Downtown and particle size distri-
butions (PSDs) measured by the Fidas OPC in Aosta–Saint Christophe. To ensure comparability between these two datasets, the particle volume distributions from the Fidas OPC (normally extending up to sizes of 18 μm) were weighted by a typical cut-off efficiency of $PM_{10}$ sampling head (Sect. S6). We then performed a PMF analysis of the hourly-averaged PSDs from the Fidas OPC (hereafter referred as size-PMF). Four principal modes were identified, as shown in Fig. 14a (their dependence on season, wind speed and direction is also provided in Fig. S9). The curves of the relative contributions of the four size-PMF
modes to the total particle volume are approximately centred at 0.2, 0.5, 2 and 10 μm diameter, respectively (and will be thus referred as 0.2, 0.5, 2 and 10-μm size-PMF modes in the following). The number of factors ($p$) was chosen based on physical considerations: if $p > 4$, then the last mode (largest sizes) splits into sub-modes, which are not relevant for the present study; conversely, if fewer components ($p < 4$) are retained, they merge into unphysical modes (e.g., very large and very fine particles combined together).

The scores from the size-PMF were then averaged on a daily basis to be compared to the chemical PMF ones (hereafter, chem-PMF; Sect. S4). Figure 14 compares time series of selected chem-PMF (dataset a) and size-PMF results. It shows that:

(a) the nitrate-rich mode from chem-PMF and the 0.5-μm size-PMF mode correlate very strongly ($\rho$=0.81, Fig. 14b);

(b) the sum of the sulfate-rich mode and traffic and heating mode correlate very strongly ($\rho$=0.82, Fig. 14c) with the 0.2-μm size-PMF mode. Note that the correlation index decreases ($\rho$ between 0.35 and 0.69) if the sulfate-rich mode and traffic
and heating mode are compared separately with the 0.2 μm mode;

(c) A moderate, statistically significant (p-value$< 2.2 \cdot 10^{-16}$), Pearson's correlation index $\rho = 0.59$, Fig. 14d) was found between the chem-PMF soil plus road salting modes and the size-PMF coarser modes (sum of 2 and 10 μm modes). This suggests that these latter likely arise from non-exhaust traffic emissions, such as abrasion from brakes, tire wear and road (with typical sizes of 2–5 μm), and resuspension from the surface (up to >10 μm), as also found in other
studies (e.g., Harrison et al., 2012). This agreement is remarkable when considering that the particles were sampled at two different sites (Aosta–Saint Christophe and Aosta–Downtown) with distinct environmental features, and that coarse particles usually travel for short ranges. It is also worth mentioning that the correlation between single modes from chem-PMF (road salting or soil) and size-PMF (2 or 10 μm) separately is weaker ($\rho$ between 0.26 and 0.55) than the





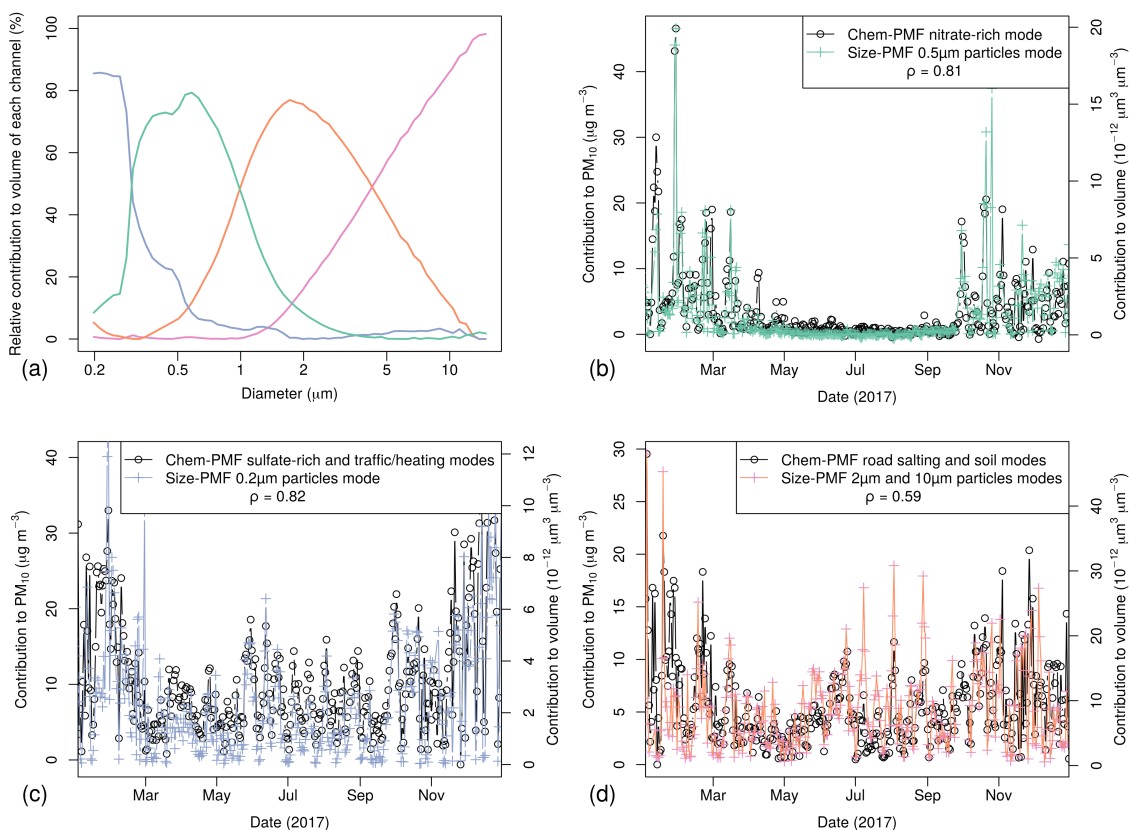

**Figure 14.** (a) Modes identified by the PMF analysis applied on the PSDs measured by the Fidas OPC (size-PMF). (b–d) Comparison between the scores obtained from PMF analysis of chemical speciation (chem-PMF on PMF-dataset a, left-hand vertical axes, mass) and size distributions (size-PMF, right-hand vertical axes, volume): (b) contributions from chem-PMF nitrate-rich and size-PMF 0.5 μm modes; (c) contributions from the sum of the chem-PMF sulfate-rich and traffic and heating modes, and from the 0.2 μm size-PMF mode; (d) sum of contributions from chem-PMF road salting and soil modes, and from 2 and 10 μm size-PMF modes.





one resulting from their sum. Finally, from a preliminary analysis based on back-trajectories and desert dust forecasts (NMMB/BSC-Dust, http://ess.bsc.es/bsc-dust-daily-forecast), the 2 μm component seems to be additionally connected to deposition, and possibly resuspension, of mineral Saharan dust in Aosta, an effect already observed in other urban areas in central Italy (Barnaba et al., 2017).

A further interesting feature is the size separation of the 0.2 and 0.5 μm accumulation modes, which likely results from different aerosol formation mechanisms in the atmosphere: condensation/coagulation from primary emissions/aging on one side (0.2-μm mode, also known as "condensation mode"), and aqueous phase processes (e.g., in fog) on the other side (forming "droplet mode" aerosol particles of about 0.5 μm diameter, e.g.,  Seinfeld and Pandis, 2006; Wang et al., 2012; Costabile et al., 2017). Finally, the very good correlation between the fine particles and the nitrate- and sulfate-rich modes at the two stations

is a further hint that this kind of particles are mostly of non-local origin. Overall, the good agreement between the size- and chem-PMF results further strengthens their independent outputs.

### 4.4   Impact of Po Valley advections on northwestern Alps columnar aerosol properties

ALC measurements revealed the vertical extent and thickness of the advected polluted layers from the Po Valley. We therefore also investigated if and how much these layers impact the column-integrated aerosol properties (i.e., those sounded by ground-

based sun photometers, but also by satellites). To this purpose, the ALC classification was applied to the measurements of the Aosta–Saint Christophe sun photometer. The average AOD at 500 nm, binned by day type and season, is shown in Fig. 15a. It can be noticed that a general increase of the AOD is found from classes A (about 0.04 on average) to F, for which AOD is more than 4 times higher (0.18). The advections are also found to affect the Ångström exponent (Fig. 15b), a parameter used to qualitatively infer mean particle size. Results from this column-integrated perspective confirm that the transported

particles are on average smaller than the locally-produced ones. In fact, the mean Ångström exponents vary from 1.0 for days A in winter and autumn, up to 1.6 for days E–F, and show a general increase among the classes for every season. To confirm and fully understand this dependence of the Ångström exponents on the ALC classification, we also investigated the columnar volume-PSDs retrieved from the sun photometer almucantar scans during the Po Valley pollution transport events. This analysis (Fig. S10) confirms what we already found with the surface-level PSDs from the Fidas OPC, i.e. that the advections increase

the number of particles in all sizes, and particularly in the sub-micron fraction. This explains the higher Ångström exponents retrieved by the sun photometer.

To again link aerosol physical and chemical properties, the variability of the single scattering albedo with day type is also shown in Fig. 15c. Yearly-averaged data are shown in this case due to the limited number of data points available for this parameter (the almucantar plane must be free of clouds to accurately retrieve the SSA, Sect. 2). Fig. 15c reveals that SSA at

500 nm generally increases from class A to class F (0.92 to 0.98), which agrees with the fact that secondary, and thus more scattering, aerosol is transported with the advections. This information is particularly important for follow-up radiative transfer evaluations under the investigated conditions. In this respect, also note that further analysis more focussed on the impact of these advection on particle absorption properties is planned for the future.





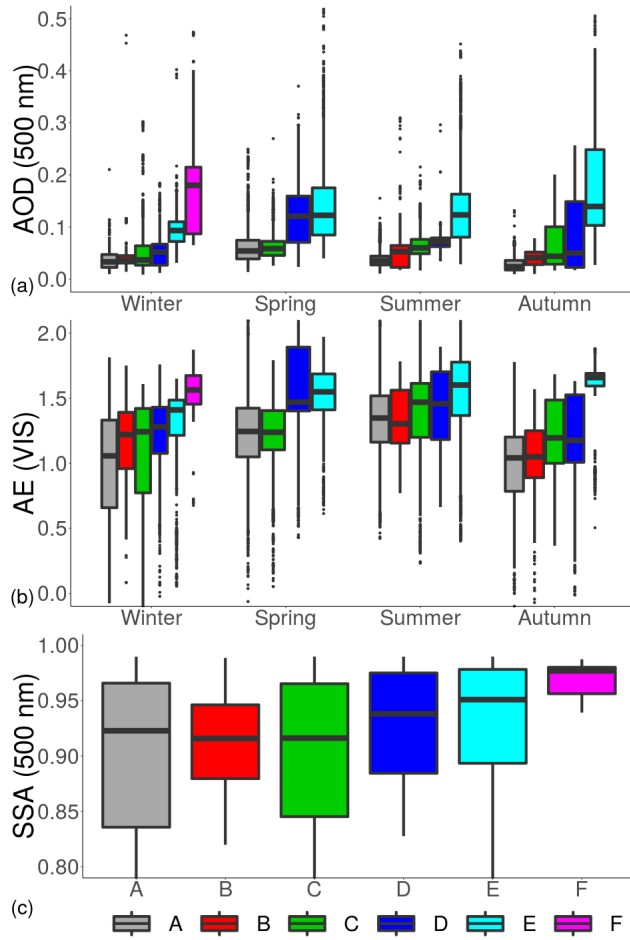

**Figure 15.** (a) Average aerosol optical depth at 500 nm from the POM sun photometer for each day type (colour code as in Fig. 12) and season; (b) Ångström exponent calculated in the 400-870 nm band; (c) Yearly-averaged single scattering albedo at 500 nm. B days in spring were removed from panels (a) and (b) due to insufficient statistics (only 3 days).

## 4.5 Comparison between observations and CTM simulations

Accurate simulations of air quality metrics are of utmost importance for Environmental Protection Agencies. Indeed, not only they are essential from a scientific/technical perspective (contributing to better understand the local and remote pollution dynamics), but they also represent fundamental duty towards the public, being air quality forecasts disseminated every day. In this section, we compare long-term (1 year, 2017) daily averages of surface PM to the corresponding simulations by FARM at two sites of the regional domain (Aosta–Downtown and Donnas). This exercise was also aimed at evaluating if and how the information gathered by the ALC can be used to understand deficiencies and thus improve the model performances. The one-year time series of both the experimental and simulated $PM_{10}$ are displayed in Fig. 16. They show similarities (such




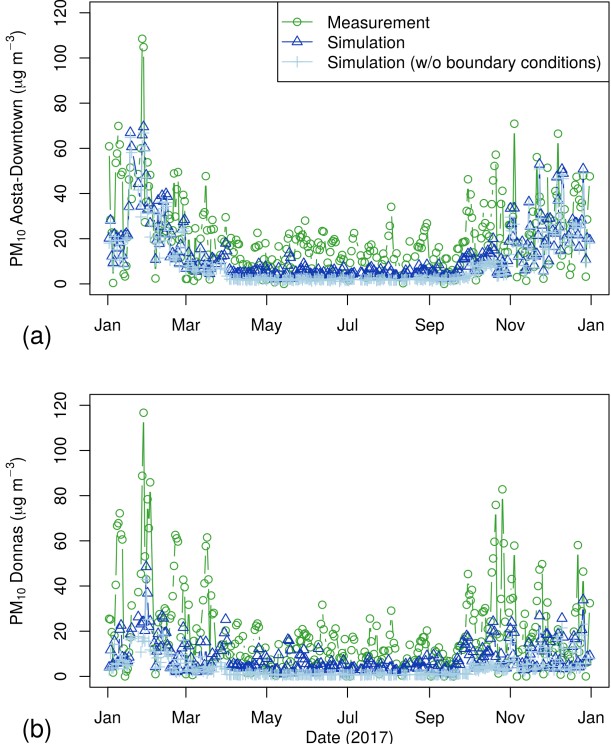

(a)

(b)

**Figure 16.** Long-term (1 year) comparison of $PM_{10}$ measurements and simulations (FARM) at the surface in Aosta–Downtown (a) and Donnas (b). Simulations including and excluding the boundary conditions are both shown.

**Table 3.** Statistics of comparison between $PM_{10}$ model forecasts and measurements at the sites of Aosta–Downtown and Donnas (mean bias error, MBE; root mean square error, RMSE; Pearson's correlation coefficient, $\rho$; normalised mean standard deviation, NMSD). Results with both $W = 1$ and $W = 4$ weighting factors to the PM fraction arriving outside the boundaries of the regional domain are reported.

| Station | W | MBE ($\mu g\,m^{-3}$) | RMSE ($\mu g\,m^{-3}$) | $\rho$ | NMSD (%) |
|---|---|---|---|---|---|
| Aosta–Downtown | 1 | -7 | 15 | 0.62 | -16 |
| Donnas | 1 | -10 | 18 | 0.49 | -59 |
| Aosta–Downtown | 4 | 1 | 13 | 0.61 | -8 |
| Donnas | 4 | 2 | 18 | 0.41 | -3 |

as the same seasonal cycle, indicating that local emissions from the regional inventory and general atmospheric processes are correctly reproduced), but also divergences (especially underestimation by FARM, as already shown and discussed in the companion paper). Table 3 summarises some statistical indicators of the model-measurements comparison, such as mean bias





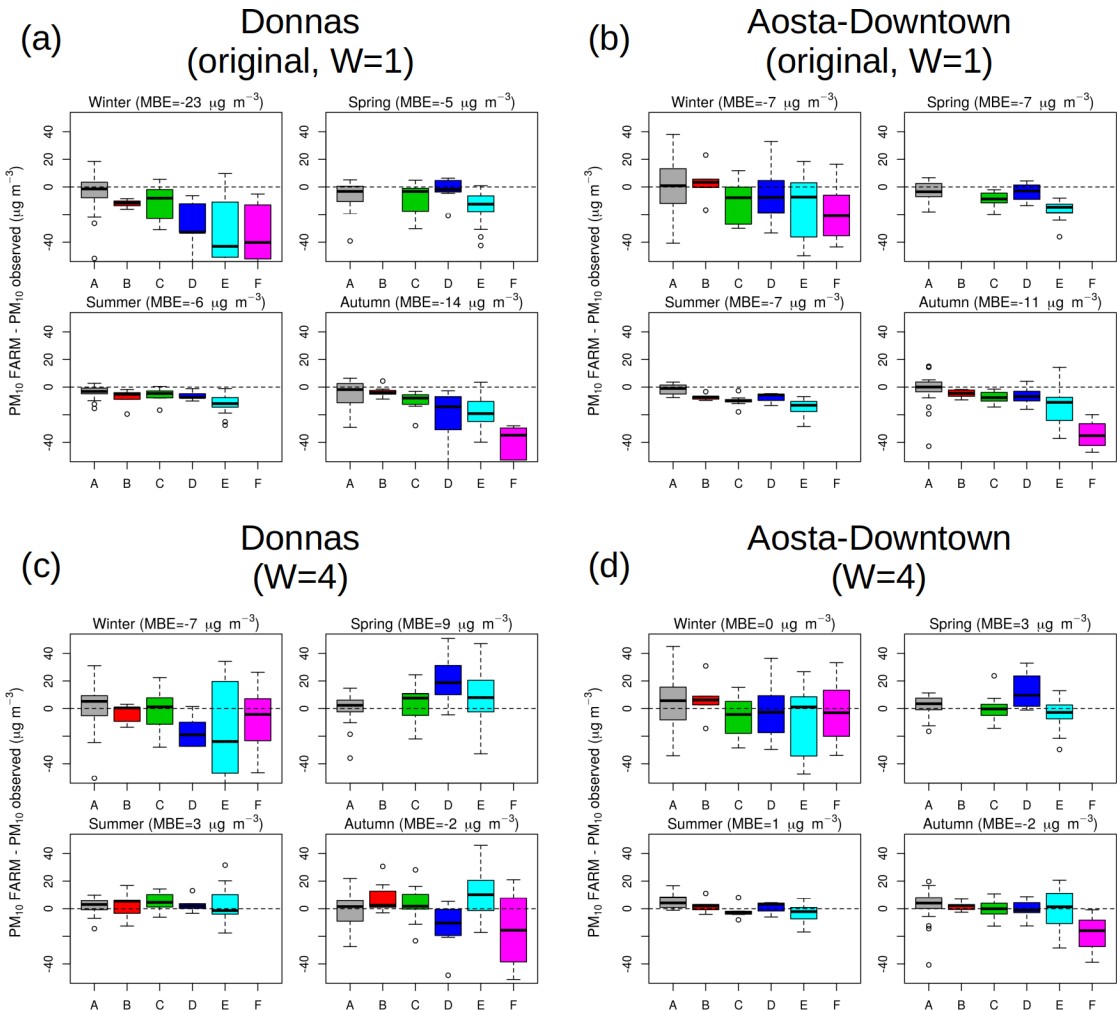

**Figure 17.** Differences between simulated and observed $PM_{10}$ concentrations at the surface. The mean bias error (MBE) for each case is reported in the plot titles. First row: FARM simulations as currently performed in ARPA for the Donnas (a) and Aosta–Downtown (b) stations. Second row: the $PM_{10}$ concentrations from outside the boundaries of the domain were multiplied by a factor $W$=4.





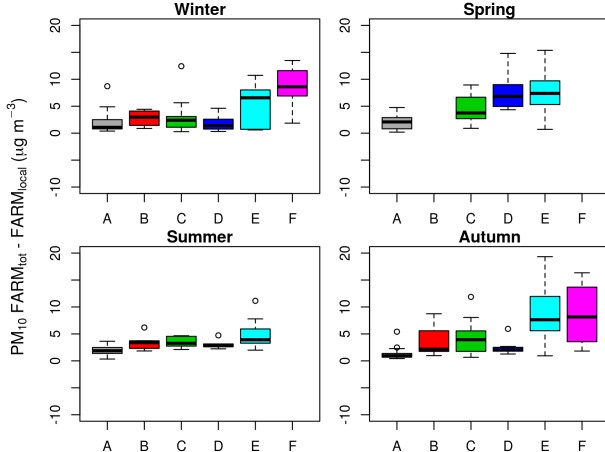

**Figure 18.** Difference between $PM_{10}$ surface concentrations in Donnas simulated by FARM taking the boundary conditions into account ($FARM_{tot}$) and without taking them into account ($FARM_{local}$, only local sources), as a function of the advection class observed by the ALC.

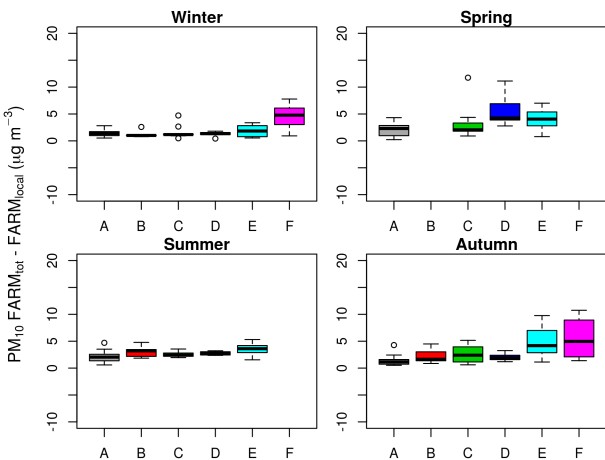

**Figure 19.** Same as Fig. 18, in Aosta–Downtown.

error (MBE), root mean square error (RMSE), Pearson's correlation coefficient ($\rho$) and normalised mean standard deviation (NMSD), i.e. ($\sigma_M/\sigma_O - 1$) (where $\sigma_M$ and $\sigma_O$ are the standard deviations of the model and observation). The overall performances of FARM are comparable to the results obtained in other studies using CTMs (e.g., Thunis et al., 2012). In Fig. 17 (a, b) we explore the differences between modelled and observed $PM_{10}$ in relation to the ALC-derived classes. It reveals that the model underestimation for both Aosta–Downtown and Donnas mostly occurs in the cases of strongest advections, with extreme model deviations as low as -40 $\mu g\,m^{-3}$ on days of class F. This is a clear indication that the external contri-





bution (boundary conditions) is not optimally parametrised in the model. We made some sensitivity tests, using a simplified approach to provide a general idea of the possible outcomes while speeding up the calculations. In particular, assuming that the contribution from outside the regional domain and the local emissions add up without interacting, we simulated the surface $PM_{10}$ concentrations turning on/off the boundary conditions (dark- and light-blue lines in Fig. 16) to roughly estimate the

contribution from sources outside the regional domain. Interestingly, the difference between the two FARM runs correlates well with the advection classes observed by the ALC (Figs. 18 and 19). This clear correlation between the simulations and the experimentally-determined atmospheric conditions indicates that the NWP model used as input to the CTM yields reasonable meteorological inputs to FARM. The reason for discrepancies is therefore likely in the emissions. To further explore the sensitivity to the boundary conditions, we then gradually increased, by a weighting factor $W$, the $PM_{10}$ concentration from

outside the boundaries of the domain trying to match the observed values. In Fig. 17(c, d) we show the results obtained with $W=4$. The overall mean bias error is much reduced compared to the original simulations, especially for the winter and autumn seasons, while slight overestimations are now visible for summer and spring. Also the annually-averaged MBE and NMSD improve (Table 3), although the other statistical indicators remain stable or even slightly worsen. Clearly, our simplified test has major limitations. For example, seasonally- and geographically-dependent (i.e., relative to the position on the regional do-

main border) factors $W$ should be used to better correct deficiencies in the national inventory. Also, the high weighting factors needed to match the measurements in winter and autumn suggest not only underestimated emissions, but also missing aerosol production mechanisms (e.g., aqueous-phase particle production as in fog/cloud processing, Gilardoni et al. (2016)) during the cold season. In fact, this simplified exercise was only intended to roughly estimate the magnitude of the "outside $PM_{10}$ tuning factor" necessary to match the observations. Indeed, this numerical experiment demonstrates that the accuracy of the air

quality forecasts may be improved by revising the inventories, principally the national one, on the basis of experimental data and particularly those from unconventional air-quality systems as ALC.

    As a last step, we use the model to extend our observational findings to a wider domain, taking at the same time into account the model performances discussed above. In Fig. 20, we provide some summarising plots of the 1-year simulations. These show the 3-D simulated fields of $PM_{10}$ concentrations in terms of both surface-level (a, b and c) and vertical transects along the main

valley (d, e and f), averaged over all non-advection and advection days in 2017 (as determined by the ALC-classification). In this latter case, simulations with $W = 1$ (b,e) and $W = 4$ (c,f) are included. During non-advection cases the aerosol is supposed to be predominantly of local origin and major hotspots are the city of Aosta and the road along the main valley (Fig. 20a). Compared to this "background condition" (ALC type A), the advection cases show higher $PM_{10}$ concentrations and a different geographical distribution, with an increase while proceeding towards the entrance of the valley (Fig. 20b, e). Obviously, the

absolute $PM_{10}$ loads are higher when the non-local contribution is multiplied by $W = 4$ (Fig. 20c), which, as discussed above, is the $W$ value allowing a better matching with the $PM_{10}$ concentrations actually measured in Aosta and Donnas. Vertical profiles of simulated concentrations are also evidently impacted by the advections (e.g., Fig. 20d–f). A more complete set of maps is provided in Fig. S11, in which contribution of particulate produced inside the regional domain (local sources) and outside the domain (boundary conditions) have been partitioned. Interesting feature in that view is that the average maps of

local $PM_{10}$ do not change between advection/non-advection cases, while the (relative) concentration maps of aerosol coming





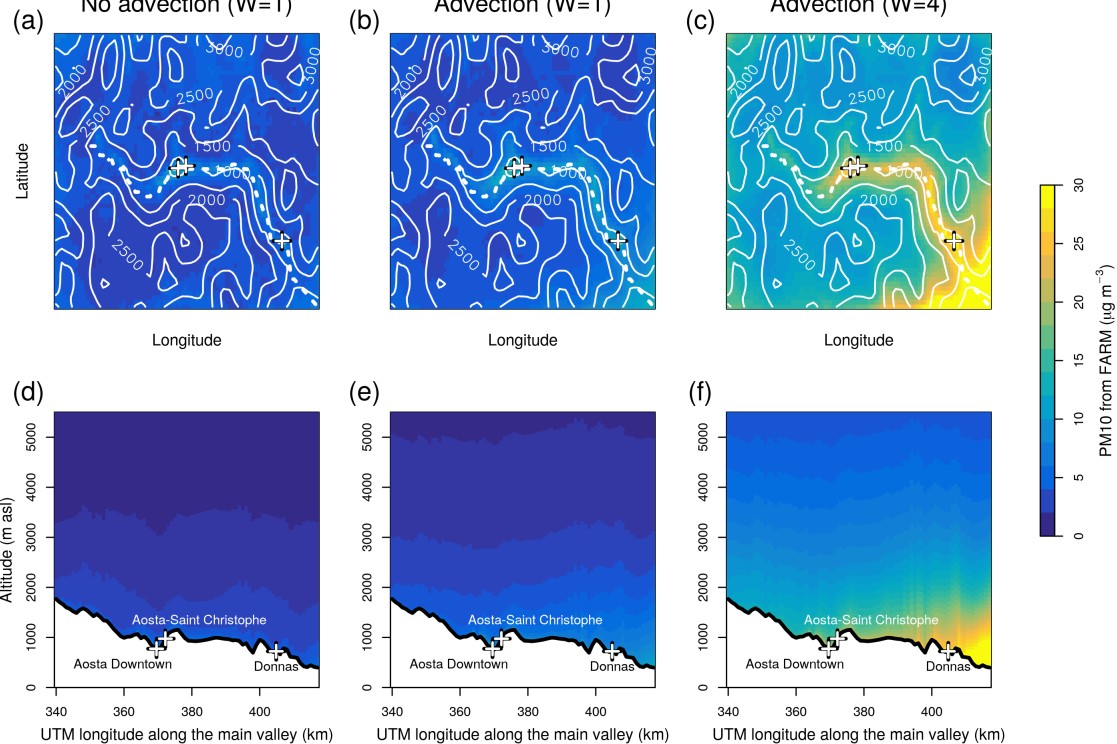

**Figure 20.** Average 3-D fields of $PM_{10}$ concentration simulated by FARM extracted at the surface level (a, b, c) and on a vertical transect (d, e, f). (a) Average of all non-advection days (categorised as "A" by the experimental ALC-classification); (b) Average of advection days ("C", "E" and "F" from ALC-classification) using a weighting factor $W = 1$ for the $PM_{10}$ contribution coming from outside the boundaries of the regional domain; (c) Same as (b) using $W = 4$ as a weighting factor. (d), (e) and (f) are vertical transects along the main valley (i.e., along the dotted line in figures (a) to (c)) corresponding to the three cases above. The altitude of the three measuring sites shown in the figures is the one from the digital elevation model, which is a smoothed representation of the real topography. The white area in the second row of figures represents the surface. The advections investigated in this study come from the east (right side of all figures).

from outside the domain show noticeable differences, thus confirming once again that the polluted air masses detected by the ALC in Aosta–Saint Christophe, on the base of which the classification is built, are of non-local origin. In conclusion, the excellent correspondence between the model output and the experimental classification based on the ALC, i.e. the remarkable difference of the concentrations and their vertical distribution between days of advection and non-advection, highlights both

5 the good performances of the CTM model and the ability of the measurement dataset used to disclose the phenomenon.

## 5 Conclusions and perspectives

In this work we used long-term (2015–2017) multi-sensor observational datasets (Table 1) coupled to modelling tools to describe pollution aerosol transport from the Po basin to the northwestern Alps. The deployment of an Automated LiDAR





Ceilometer (ALC) with its operational (24/7) capabilities in Aosta allowed: a) to collect and present the first-long term dataset of aerosol vertical distribution in the northwestern Alps; b) to provide observational evidence of the complex structure and dynamics of the atmosphere in the Alpine valleys; c) to stimulate the investigation of the phenomenon on a 4-D scale with complementing observational and modelling tools. The analysis over the long-term allowed to complement the investigation of specific case studies provided in the companion paper and answer some key scientific questions as listed in the Introduction:

1. Driving meteorological factors and frequency of the aerosol pollution transport events. The newly-developed classification based on the ALC profiles (928 classified days, Fig. 3) permitted us to aggregate the days with similar aerosol vertical structures and examine the average properties of each group. Notably, we could understand the link between the wind flow regimes and the aerosol transport. The frequency of the elevated layers was observed to be on average 43–50% of the days, depending on season (i.e. slightly less than 60% of the days falling in an ALC class different from "Other" in winter and spring to more than 70% in summer) and is clearly connected to eastern wind flows, such as plain-to-mountain, thermally-driven winds (blowing nearly 70% of the days in summer, Fig. 4). This was even clearer using Trajectory Statistical Models (Fig. 13) and contingency tables (Fig. 5) showing the clear connections between the arrival of an elevated aerosol layer over the northwestern Alps and the wind provenance. Such layers were observed 93% of the days with easterly winds (Fig. 5a), with increasing probability of occurrence for increasing air masses residence time over the Po plain (aerosol layer detected >80% of days when the trajectory residence times in the Po basin exceed 1 h, and up to 100% for residence times > 10 h, Fig. 5b);

2. Advected aerosol physical and chemical properties. The general spatio-temporal characteristics of the aerosol layers were statistically assessed based on the multi-year ALC series. The top altitudes of the layer ranges from 500 m to 4000 m a.g.l., depending on season and according to the convective boundary layer height, and the duration of the phenomenon ranges from few hours to several days in cases of atmospheric stability. The mean optical and microphysical properties of the aerosol layers were further sounded using a sun photometer. This showed that the columnar aerosol properties are all impacted by the pollution transport: aerosol optical depth (AOD at 500 nm) increases from 0.04 on non-event days up to 0.18 on event days, on average; relevant values of Ångström exponent and single scattering albedo (SSA at 500 nm) change from 1.0 to 1.6 and from 0.92 to 0.98, respectively, and depend on the duration/severity of the advection (Fig. 15). This indicates that the transported particles are on average smaller and more light-scattering than the locally-produced ones and agrees well with the results of measurements and analyses of aerosol properties at the surface. Positive matrix factorisation (PMF) of chemical properties at surface level allowed to disclose that particles advected from the Po Valley to the northwestern Alps are of secondary origin (e.g., Fig. 12) and mainly composed by nitrates, sulfates and ammonium. Interestingly, we also found an organic component in the warm season, which confirms the Po Valley as an OM source. Moreover, particle size distributions were observed to increase in the fine ($< 1\ \mu m$) fraction during event days. Correlation of chemical-PMF with independent PMF performed on aerosol size distribution also provided evidence of a clear link between chemical properties and aerosol size (correlations indexes $\rho > 0.8$, Fig. 14), which proves that different aerosol formation mechanisms act in the atmosphere in different seasons. In particular, we found some evidence





of aqueous processing of particles in winter (e.g., fog and/or cloud processing) through identification of a PMF "droplet mode" in the winter results;

3. Aerosol transport impact on air quality. At the surface level, the advections were demonstrated to alter both the absolute concentrations of $PM_{10}$ (these being up to 3.5 times higher during event days compared to non-event days, Fig. 8), and their daily cycles, with hourly variations up to $30\,\mu g\,m^{-3}$ (Fig. 10). PMF statistics on chemical data identified 5 to 7 different sources, depending on the available chemical characterisation, two of which (nitrate-rich mode in winter and sulfate-rich mode in summer) were attributed to the arrival of particles from outside the Aosta Valley as also confirmed by Trajectory Statistical Models (Fig. 13). Using their relevant scores as a proxy, we estimate an average 25% contribution of these transported nitrate and sulfate-rich components to the Aosta $PM_{10}$, reaching values of $50\,\mu g\,m^{-3}$ in some episodes and, more generally, favouring exceedances in proximity of the EU legislative limits. This non-local relative contribution is expected to be even much larger in rural and remote sites of the region. Increasing the number of stations collecting samples for chemical analyses would be an important follow-up to estimate the non-local contribution to $PM_{10}$ in non-urban sites. The good correlation found between the $PM_{10}$ concentration anomalies in the urban site of Aosta–Downtown and in the regional site of Donnas ($\rho > 0.7$, Fig. 9) is however a clear indication that aerosol loads all over the region are mainly influenced by trans-regional transport rather than by local emissions. It is also worth mentioning that the aerosol particle number (here only measured in the 0.18–18 µm range, i.e., missing the finest particles) increased remarkably (up to $>3000\,particles\,cm^{-3}$) in some transport episodes, with possible implications on human health. Some preliminary results also indicate that these regular air mass transport is also able to alter the oxidative capacity of the atmosphere ($NO_x/NO$ ratio), although further investigation is needed to better quantify it;

4. Ability to predict the arrival and impacts of polluted air mass transport. Experimental data and their coupling with modelling tools well demonstrated the Po plain origin of the polluted layers and their increased probability of detection as a function of the air masses residence time over the origin region. Current chemical transport models, however, were found to reproduce well the phenomenon in qualitative terms, but to generally underestimate the absolute advected $PM_{10}$. Based on a 1-year long simulated dataset, we tested how the air quality forecasts could be improved by updating the emission inventories and using the ALC to constrain the modelled emissions. After the tested measurements-based tuning of the national inventory ("boundary conditions" increased by a factor of 4), the model underestimation was reduced from biases $< -40\,\mu g\,m^{-3}$ to $-20\,\mu g\,m^{-3}$ in the worst cases, with mean average biases $< 2\,\mu g\,m^{-3}$ (Table 3), thus enhancing our ability to correctly predict the PM concentration and their relevant impacts (Fig. 20). Still, further improvements are necessary to enhance the model ability to reproduce aerosol processes in aqueous-phase chemistry (e.g., Gong et al., 2011).

In conclusion, we demonstrated that, despite considered a pristine environment, the northwestern Alps are regularly affected by a sort of "polluted aerosol tide", i.e. by a wind-driven transport of polluted aerosol layers from the Po plain. Overall, this calls for air pollution mitigation policies acting, at least, over the regional scale in this fragile environment.



*Data availability.* The ALC data are available upon request from the Alice-net (alicenet@isac.cnr.it) and E-PROFILE (http://data.ceda. ac.uk/badc/eprofile/data/) networks. The sun photometer data can be downloaded from the EuroSkyRad network web site (http://www. euroskyrad.net/index.html) after authentication (credentials may be requested to M. Campanelli, m.campanelli@isac.cnr.it). The measurements from ARPA air quality surface network are available at the web page http://www.arpa.vda.it/it/aria/la-qualit%C3%A0-dell-aria/

stazioni-di-monitoraggio/inquinanti-export-dati. The weather data from the Aosta–Saint Christophe station can be retrieved from http: //cf.regione.vda.it/richiesta_dati.php upon request to Centro Funzionale della Valle d'Aosta. The rest of the data can be asked to the corresponding author (h.diemoz@arpa.vda.it).

*Author contributions.* HD, FB and GPG conceived and designed the study, interpreted the results and wrote the paper. HD analysed the data. TM supplied the meteorological observations and numerical weather predictions. GP performed the chemical transport simulations.

SP carried out the EC/OC analyses and IKFT helped with the interpretation of the chemical speciation. MC supplied the POM calibration factors.

*Competing interests.* The authors declare that they have no conflict of interest.

*Acknowledgements.* The authors would like to thank: A. Brunier, G. Lupato, P. Proment, S. Vaccari, and M.C. Gibellino (ARPA Valle d'Aosta) for carrying out the chemical analyses; M. Pignet, C. Tarricone, and M. Zublena (ARPA Valle d'Aosta) for providing the data from

the air quality surface network; P. Lazzeri (APPA Trento) for helping with the PMF analysis. The authors would like to acknowledge the valuable contribution of the discussions in the working group meetings organised by COST Action ES1303 (TOPROF). They also gratefully acknowledge the Institute for Atmospheric and Climate Science, ETH Zurich, Switzerland for the provision of the LAGRANTO software used in this publication.





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

©c Author(s) 2019. CC BY 4.0 License.





Carter, W.: Documentation of the SAPRC-99 Chemical Mechanism for VOC Reactivity Assessment. Final Report to California Air Resources Board, Contract 92-329 and 95-308, SAPRC, Tech. rep., University of California, Riverside,CA, http://www.engr.ucr.edu/~carter/pubs/s99doc.pdf, 2000.

Cavalli, F., Viana, M., Yttri, K. E., Genberg, J., and Putaud, J.-P.: Toward a standardised thermal-optical protocol for measuring atmospheric organic and elemental carbon: the EUSAAR protocol, Atmos. Meas. Tech., 3, 79–89, https://doi.org/10.5194/amt-3-79-2010, 2010.

Cesaroni, G., Badaloni, C., Gariazzo, C., Stafoggia, M., Sozzi, R., Davoli, M., and Forastiere, F.: Long-term exposure to urban air pollution and mortality in a cohort of more than a million adults in Rome, Environ. Health Persp., 121, 324, https://doi.org/10.1289/ehp.1205862, 2013.

Charron, A., Harrison, R. M., Moorcroft, S., and Booker, J.: Quantitative interpretation of divergence between PM10 and PM2.5 mass measurement by TEOM and gravimetric (Partisol) instruments, Atmos. Environ., 38, 415 – 423, https://doi.org/10.1016/j.atmosenv.2003.09.072, 2004.

Chazette, P., Couvert, P., Randriamiarisoa, H., Sanak, J., Bonsang, B., Moral, P., Berthier, S., Salanave, S., and Toussaint, F.: Three-dimensional survey of pollution during winter in French Alps valleys, Atmos. Environ., 39, 1035 – 1047, https://doi.org/https://doi.org/10.1016/j.atmosenv.2004.10.014, 2005.

Chemel, C., Arduini, G., Staquet, C., Largeron, Y., Legain, D., Tzanos, D., and Paci, A.: Valley heat deficit as a bulk measure of wintertime particulate air pollution in the Arve River Valley, Atmos. Environ., 128, 208 – 215, https://doi.org/10.1016/j.atmosenv.2015.12.058, 2016.

Chu, D. A., Kaufman, Y. J., Zibordi, G., Chern, J. D., Mao, J., Li, C., and Holben, B. N.: Global monitoring of air pollution over land from the Earth Observing System-Terra Moderate Resolution Imaging Spectroradiometer (MODIS), J. Geophys. Res., 108, https://doi.org/10.1029/2002JD003179, 2003.

Clerici, M. and Mélin, F.: Aerosol direct radiative effect in the Po Valley region derived from AERONET measurements, Atmos. Chem. Phys., 8, 4925–4946, https://doi.org/10.5194/acp-8-4925-2008, 2008.

Cong, Z., Kawamura, K., Kang, S., and Fu, P.: Penetration of biomass-burning emissions from South Asia through the Himalayas: new insights from atmospheric organic acids, Scientific reports, 5, https://doi.org/10.1038/srep09580, 2015.

Costabile, F., Gilardoni, S., Barnaba, F., Di Ianni, A., Di Liberto, L., Dionisi, D., Manigrasso, M., Paglione, M., Poluzzi, V., Rinaldi, M., Facchini, M. C., and Gobbi, G. P.: Characteristics of brown carbon in the urban Po Valley atmosphere, Atmospheric Chemistry and Physics, 17, 313–326, https://doi.org/10.5194/acp-17-313-2017, https://www.atmos-chem-phys.net/17/313/2017/, 2017.

Cugerone, K., De Michele, C., Ghezzi, A., Gianelle, V., and Gilardoni, S.: On the functional form of particle number size distributions: influence of particle source and meteorological variables, Atmos. Chem. Phys., 18, 4831–4842, https://doi.org/10.5194/acp-18-4831-2018, 2018.

Curci, G., Ferrero, L., Tuccella, P., Barnaba, F., Angelini, F., Bolzacchini, E., Carbone, C., Denier van der Gon, H. A. C., Facchini, M. C., Gobbi, G. P., Kuenen, J. P. P., Landi, T. C., Perrino, C., Perrone, M. G., Sangiorgi, G., and Stocchi, P.: How much is particulate matter near the ground influenced by upper-level processes within and above the PBL? A summertime case study in Milan (Italy) evidences the distinctive role of nitrate, Atmos. Chem. Phys., 15, 2629–2649, https://doi.org/10.5194/acp-15-2629-2015, 2015.

de Freitas, C. R.: Tourism climatology: evaluating environmental information for decision making and business planning in the recreation and tourism sector, Int. J. Biometeorol., 48, 45–54, https://doi.org/10.1007/s00484-003-0177-z, 2003.

Decesari, S., Allan, J., Plass-Duelmer, C., Williams, B. J., Paglione, M., Facchini, M. C., O'Dowd, C., Harrison, R. M., Gietl, J. K., Coe, H., Giulianelli, L., Gobbi, G. P., Lanconelli, C., Carbone, C., Worsnop, D., Lambe, A. T., Ahern, A. T., Moretti, F., Tagliavini, E., Elste,





T., Gilge, S., Zhang, Y., and Dall'Osto, M.: Measurements of the aerosol chemical composition and mixing state in the Po Valley using multiple spectroscopic techniques, Atmos. Chem. Phys., 14, 12 109–12 132, https://doi.org/10.5194/acp-14-12109-2014, 2014.

Dhungel, S., Kathayat, B., Mahata, K., and Panday, A.: Transport of regional pollutants through a remote trans-Himalayan valley in Nepal, Atmos. Chem. Phys., 18, 1203–1216, https://doi.org/10.5194/acp-18-1203-2018, 2018.

Diémoz, H., Siani, A. M., Casale, G. R., di Sarra, A., Serpillo, B., Petkov, B., Scaglione, S., Bonino, A., Facta, S., Fedele, F., Grifoni, D., Verdi, L., and Zipoli, G.: First national intercomparison of solar ultraviolet radiometers in Italy, Atmos. Meas. Tech., 4, 1689–1703, https://doi.org/10.5194/amt-4-1689-2011, 2011.

Diémoz, H., Campanelli, M., and Estellés, V.: One Year of Measurements with a POM-02 Sky Radiometer at an Alpine EuroSkyRad Station, J. Meteorol. Soc. Jpn., 92A, 1–16, https://doi.org/10.2151/jmsj.2014-A01, 2014a.

Diémoz, H., Siani, A. M., Redondas, A., Savastiouk, V., McElroy, C. T., Navarro-Comas, M., and Hase, F.: Improved retrieval of nitrogen dioxide ($NO_2$) column densities by means of MKIV Brewer spectrophotometers, Atmos. Meas. Tech., 7, 4009–4022, https://doi.org/10.5194/amt-7-4009-2014, 2014b.

Diémoz, H., Barnaba, F., Magri, T., Pession, G., Dionisi, D., Pittavino, S., Tombolato, I. K. F., Campanelli, M., Della Ceca, L. S., Hervo, M., Di Liberto, L., Ferrero, L., and Gobbi, G. P.: Transport of Po Valley aerosol pollution to the northwestern Alps – Part 1: Phenomenology,

Atmos. Chem. Phys., 19, 3065–3095, https://doi.org/10.5194/acp-19-3065-2019, 2019.

Dionisi, D., Barnaba, F., Diémoz, H., Di Liberto, L., and Gobbi, G. P.: A multiwavelength numerical model in support of quantitative retrievals of aerosol properties from automated lidar ceilometers and test applications for AOT and $PM_{10}$ estimation, Atmos. Meas. Tech., 11, 6013–6042, https://doi.org/10.5194/amt-11-6013-2018, 2018.

Dosio, A., Galmarini, S., and Graziani, G.: Simulation of the circulation and related photochemical ozone dispersion in the Po

plains (northern Italy): Comparison with the observations of a measuring campaign, J. Geophys. Res., 107, LOP 2–1–LOP 2–24, https://doi.org/10.1029/2000JD000046, 2002.

EEA: Air Quality in Europe - 2015 Report, Tech. rep., https://doi.org/10.2800/62459, 2015.

EEA: Air Quality in Europe - 2017 Report, Tech. rep., https://doi.org/10.2800/850018, 2017.

Egger, J., Bajrachaya, S., Egger, U., Heinrich, R., Reuder, J., Shayka, P., Wendt, H., and Wirth, V.: Diurnal Winds in

the Himalayan Kali Gandaki Valley. Part I: Observations, Mon. Weather Rev., 128, 1106–1122, https://doi.org/10.1175/1520-0493(2000)128<1106:DWITHK>2.0.CO;2, 2000.

EMEP: Transboundary particulate matter, photo-oxidants, acidifying and eutrophying components, Tech. rep., MSC-W, CCC and CEIP, 2016.

EU Commission: Directive 2008/50/EC of the European Parliament and of the Council of 21 May 2008 on ambient air quality and cleaner

air for Europe, Official Journal of the European Union, pp. L152/1–44, 2008.

EU Commission: European Commission – Press release: Air quality: Commission takes action to protect citizens from air pollution, Brussels, 17 May 2018, Press Release Database, http://europa.eu/rapid/press-release_IP-18-3450_en.htm, 2018.

Fernald, F. G.: Analysis of atmospheric lidar observations: some comments, Appl. Opt., 23, 652–653, https://doi.org/10.1364/AO.23.000652, 1984.

Ferrero, L., Perrone, M. G., Petraccone, S., Sangiorgi, G., Ferrini, B. S., Lo Porto, C., Lazzati, Z., Cocchi, D., Bruno, F., Greco, F., Riccio, A., and Bolzacchini, E.: Vertically-resolved particle size distribution within and above the mixing layer over the Milan metropolitan area, Atmos. Chem. Phys., 10, 3915–3932, https://doi.org/10.5194/acp-10-3915-2010, 2010.





Ferrero, L., Castelli, M., Ferrini, B. S., Moscatelli, M., Perrone, M. G., Sangiorgi, G., D'Angelo, L., Rovelli, G., Moroni, B., Scardazza, F., Močnik, G., Bolzacchini, E., Petitta, M., and Cappelletti, D.: Impact of black carbon aerosol over Italian basin valleys: high-resolution measurements along vertical profiles, radiative forcing and heating rate, Atmos. Chem. Phys., 14, 9641–9664, https://doi.org/10.5194/acp-14-9641-2014, 2014.

Finardi, S., Silibello, C., D'Allura, A., and Radice, P.: Analysis of pollutants exchange between the Po Valley and the surrounding European region, Urban Clim., 10, 682 – 702, https://doi.org/https://doi.org/10.1016/j.uclim.2014.02.002, source apportionment and modelling of urban air pollution, 2014.

Fuzzi, S., Baltensperger, U., Carslaw, K., Decesari, S., Denier van der Gon, H., Facchini, M. C., Fowler, D., Koren, I., Langford, B., Lohmann, U., Nemitz, E., Pandis, S., Riipinen, I., Rudich, Y., Schaap, M., Slowik, J. G., Spracklen, D. V., Vignati, E., Wild, M., Williams, M.,
and Gilardoni, S.: Particulate matter, air quality and climate: lessons learned and future needs, Atmos. Chem. Phys., 15, 8217–8299, https://doi.org/10.5194/acp-15-8217-2015, 2015.

Gariazzo, C., Silibello, C., Finardi, S., Radice, P., Piersanti, A., Calori, G., Cecinato, A., Perrino, C., Nussio, F., Cagnoli, M., et al.: A gas/aerosol air pollutants study over the urban area of Rome using a comprehensive chemical transport model, Atmos. Environ., 41, 7286–7303, https://doi.org/10.1016/j.atmosenv.2007.05.018, 2007.

Gilardoni, S., Vignati, E., Cavalli, F., Putaud, J. P., Larsen, B. R., Karl, M., Stenström, K., Genberg, J., Henne, S., and Dentener, F.: Better constraints on sources of carbonaceous aerosols using a combined $^{14}$C – macro tracer analysis in a European rural background site, Atmos. Chem. Phys., 11, 5685–5700, https://doi.org/10.5194/acp-11-5685-2011, 2011.

Gilardoni, S., Massoli, P., Giulianelli, L., Rinaldi, M., Paglione, M., Pollini, F., Lanconelli, C., Poluzzi, V., Carbone, S., Hillamo, R., Russell, L. M., Facchini, M. C., and Fuzzi, S.: Fog scavenging of organic and inorganic aerosol in the Po Valley, Atmos. Chem. Phys., 14, 6967–
6981, https://doi.org/10.5194/acp-14-6967-2014, 2014.

Gilardoni, S., Massoli, P., Paglione, M., Giulianelli, L., Carbone, C., Rinaldi, M., Decesari, S., Sandrini, S., Costabile, F., Gobbi, G. P., Pietrogrande, M. C., Visentin, M., Scotto, F., Fuzzi, S., and Facchini, M. C.: Direct observation of aqueous secondary organic aerosol from biomass-burning emissions, P. Natl. A. Sci., 113, 10 013–10 018, https://doi.org/10.1073/pnas.1602212113, 2016.

Giovannini, L., Laiti, L., Serafin, S., and Zardi, D.: The thermally driven diurnal wind system of the Adige Valley in the Italian Alps, Q. J.
Roy. Meteor. Soc., 143, 2389–2402, https://doi.org/10.1002/qj.3092, 2017.

Gohm, A., Harnisch, F., Vergeiner, J., Obleitner, F., Schnitzhofer, R., Hansel, A., Fix, A., Neininger, B., Emeis, S., and Schäfer, K.: Air Pollution Transport in an Alpine Valley: Results From Airborne and Ground-Based Observations, Bound.-Lay. Meteorol., 131, 441–463, https://doi.org/10.1007/s10546-009-9371-9, 2009.

Gong, W., Stroud, C., and Zhang, L.: Cloud Processing of Gases and Aerosols in Air Quality Modeling, Atmosphere, 2, 567–616,
https://doi.org/10.3390/atmos2040567, 2011.

Green, D. C., Fuller, G. W., and Baker, T.: Development and validation of the volatile correction model for PM10 – An empirical method for adjusting TEOM measurements for their loss of volatile particulate matter, Atmos. Environ., 43, 2132 – 2141, https://doi.org/10.1016/j.atmosenv.2009.01.024, 2009.

Hann, J. v.: Zur Theorie der Berg- und Talwinde, Z. Öst. Meteorol., 14, 444–448, 1879.

Harrison, R. M., Jones, A. M., Gietl, J., Yin, J., and Green, D. C.: Estimation of the Contributions of Brake Dust, Tire Wear, and Resuspension to Nonexhaust Traffic Particles Derived from Atmospheric Measurements, Environ. Sci. Tech., 46, 6523–6529, https://doi.org/10.1021/es300894r, 2012.



Hashimoto, M., Nakajima, T., Dubovik, O., Campanelli, M., Che, H., Khatri, P., Takamura, T., and Pandithurai, G.: Development of a new data-processing method for SKYNET sky radiometer observations, Atmos. Meas. Tech., 5, 2723–2737, https://doi.org/10.5194/amt-5-2723-2012, 2012.

Hsu, Y.-K., Holsen, T. M., and Hopke, P. K.: Comparison of hybrid receptor models to locate PCB sources in Chicago, Atmos. Environ., 37, 545 – 562, https://doi.org/https://doi.org/10.1016/S1352-2310(02)00886-5, 2003.

Kabashnikov, V. P., Chaikovsky, Anatoli, P., Kucsera, T. L., and Metelskaya, N. S.: Estimated accuracy of three common trajectory statistical methods, Atmos. Environ., 45, 5425 – 5430, https://doi.org/https://doi.org/10.1016/j.atmosenv.2011.07.006, 2011.

Kaiser, A.: Origin of polluted air masses in the Alps. An overview and first results for MONARPOP, Environ. Pollut., 157, 3232 – 3237, https://doi.org/https://doi.org/10.1016/j.envpol.2009.05.042, 2009.

Kambezidis, H. and Kaskaoutis, D.: Aerosol climatology over four AERONET sites: An overview, Atmos. Environ., 42, 1892 – 1906, https://doi.org/10.1016/j.atmosenv.2007.11.013, 2008.

Kastendeuch, P. P. and Kaufmann, P.: Classification of summer wind fields over complex terrain, Int. J. Climatol., 17, 521–534, https://doi.org/10.1002/(SICI)1097-0088(199704)17:5<521::AID-JOC143>3.0.CO;2-Q, 1997.

Kazadzis, S., Kouremeti, N., Diémoz, H., Gröbner, J., Forgan, B. W., Campanelli, M., Estellés, V., Lantz, K., Michalsky, J., Carlund, T., Cuevas, E., Toledano, C., Becker, R., Nyeki, S., Kosmopoulos, P. G., Tatsiankou, V., Vuilleumier, L., Denn, F. M., Ohkawara, N., Ijima, O., Goloub, P., Raptis, P. I., Milner, M., Behrens, K., Barreto, A., Martucci, G., Hall, E., Wendell, J., Fabbri, B. E., and Wehrli, C.: Results from the Fourth WMO Filter Radiometer Comparison for aerosol optical depth measurements, Atmos. Chem. Phys., 18, 3185–3201, https://doi.org/10.5194/acp-18-3185-2018, 2018.

Keiser, D., Lade, G., and Rudik, I.: Air pollution and visitation at U.S. national parks, Sci. Adv., 4, https://doi.org/10.1126/sciadv.aat1613, 2018.

Khan, M., Masiol, M., Formenton, G., Di Gilio, A., de Gennaro, G., Agostinelli, C., and Pavoni, B.: Carbonaceous PM2.5 and secondary organic aerosol across the Veneto region (NE Italy), Sci. Total Environ., 542, 172 – 181, https://doi.org/https://doi.org/10.1016/j.scitotenv.2015.10.103, 2016.

Khatri, P. and Takamura, T.: An Algorithm to Screen Cloud-Affected Data for Sky Radiometer Data Analysis, J. Meteorol. Soc. Jpn., 87, 189–204, https://doi.org/10.2151/jmsj.87.189, 2009.

Klett, J. D.: Lidar inversion with variable backscatter/extinction ratios, Appl. Opt., 24, 1638–1643, https://doi.org/10.1364/AO.24.001638, 1985.

Kukkonen, J., Sokhi, R., Luhana, L., Härkönen, J., Salmi, T., Sofiev, M., and Karppinen, A.: Evaluation and application of a statistical model for assessment of long-range transported proportion of PM2.5 in the United Kingdom and in Finland, Atmos. Environ., 42, 3980 – 3991, https://doi.org/10.1016/j.atmosenv.2007.02.036, fifth International Conference on Urban Air Quality, 2008.

Landi, T., Curci, G., Carbone, C., Menut, L., Bessagnet, B., Giulianelli, L., Paglione, M., and Facchini, M.: Simulation of size-segregated aerosol chemical composition over northern Italy in clear sky and wind calm conditions, Atmos. Res., 125-126, 1 – 11, https://doi.org/https://doi.org/10.1016/j.atmosres.2013.01.009, 2013.

Largeron, Y. and Staquet, C.: Persistent inversion dynamics and wintertime PM10 air pollution in Alpine valleys, Atmos. Environ., 135, 92 – 108, https://doi.org/10.1016/j.atmosenv.2016.03.045, 2016.

Larsen, B., Gilardoni, S., Stenström, K., Niedzialek, J., Jimenez, J., and Belis, C.: Sources for PM air pollution in the Po Plain, Italy: II. Probabilistic uncertainty characterization and sensitivity analysis of secondary and primary sources, Atmos. Environ., 50, 203 – 213, https://doi.org/https://doi.org/10.1016/j.atmosenv.2011.12.038, 2012.




Loomis, D., Grosse, Y., Lauby-Secretan, B., El Ghissassi, F., Bouvard, V., Benbrahim-Tallaa, L., Guha, N., Baan, R., Mattock, H., and Straif, K.: The carcinogenicity of outdoor air pollution, Lancet Oncology, 14, 1262, https://doi.org/10.1016/S1470-2045(13)70487-X show, 2013.

Mann, H. B. and Whitney, D. R.: On a Test of Whether one of Two Random Variables is Stochastically Larger than the Other, Ann. Math. Stat., 18, 50–60, 1947.

Matta, E., Facchini, M. C., Decesari, S., Mircea, M., Cavalli, F., Fuzzi, S., Putaud, J.-P., and Dell'Acqua, A.: Mass closure on the chemical species in size-segregated atmospheric aerosol collected in an urban area of the Po Valley, Italy, Atmos. Chem. Phys., 3, 623–637, https://doi.org/10.5194/acp-3-623-2003, 2003.

Mazzola, M., Lanconelli, C., Lupi, A., Busetto, M., Vitale, V., and Tomasi, C.: Columnar aerosol optical properties in the Po Valley, Italy, from MFRSR data, J. Geophys. Res., 115, https://doi.org/10.1029/2009JD013310, 2010.

Mélin, F. and Zibordi, G.: Aerosol variability in the Po Valley analyzed from automated optical measurements, Geophy. Res. Lett., 32, https://doi.org/10.1029/2004GL021787, 2005.

Norris, G. and Duvall, R.: EPA Positive Matrix Factorization (PMF) 5.0 – Fundamentals and User Guide, U.S. Environmental Protection Agency, https://www.epa.gov/sites/production/files/2015-02/documents/pmf_5.0_user_guide.pdf, 2014.

Nyeki, S., Eleftheriadis, K., Baltensperger, U., Colbeck, I., Fiebig, M., Fix, A., Kiemle, C., Lazaridis, M., and Petzold, A.: Airborne Lidar and

in-situ Aerosol Observations of an Elevated Layer, Leeward of the European Alps and Apennines, Geophys. Res. Lett., 29, 33–1–33–4, https://doi.org/10.1029/2002GL014897, 2002.

Paatero, P.: Least squares formulation of robust non-negative factor analysis, Chemometr. Intell. Lab., 37, 23 – 35, https://doi.org/10.1016/S0169-7439(96)00044-5, 1997.

Paatero, P. and Tapper, U.: Positive matrix factorization: A non-negative factor model with optimal utilization of error estimates of data

values, Environmetrics, 5, 111–126, https://doi.org/10.1002/env.3170050203, 1994.

Patashnick, H. and Rupprecht, E. G.: Continuous PM-10 Measurements Using the Tapered Element Oscillating Microbalance, J. Air Waste Manage., 41, 1079–1083, https://doi.org/10.1080/10473289.1991.10466903, 1991.

Pepin, N., Bradley, R. S., Diaz, H. F., Baraer, M., Caceres, E. B., Forsythe, N., Fowler, H., Greenwood, G., Hashmi, M. Z., Liu, X. D., Miller, J. R., Ning, L., Ohmura, A., Palazzi, E., Rangwala, I., Schöner, W., Severskiy, I., Shahgedanova, M., Wang, M. B.,

Williamson, S. N., and Yang, D. Q.: Elevation-dependent warming in mountain regions of the world, Nat. Clim. Change, 5, 424–430, https://doi.org/10.1038/nclimate2563, 2015.

Perrone, M., Larsen, B., Ferrero, L., Sangiorgi, G., De Gennaro, G., Udisti, R., Zangrando, R., Gambaro, A., and Bolzacchini, E.: Sources of high PM2.5 concentrations in Milan, Northern Italy: Molecular marker data and CMB modelling, Sci. Total Environ., 414, 343 – 355, https://doi.org/10.1016/j.scitotenv.2011.11.026, 2012.

Philipona, R.: Greenhouse warming and solar brightening in and around the Alps, Int. J. Climatol., 33, 1530–1537, https://doi.org/10.1002/joc.3531, 2013.

Pletscher, K., Weiss, M., and Moelter, L.: Simultaneous determination of PM fractions, particle number and particle size distribution in high time resolution applying one and the same optical measurement technique, Gefahrst. Reinhalt. L., 76, 425–436, http://www.gefahrstoffe.de/gest/article.php?data[article_id]=86622, 2016.

Putaud, J. P., Van Dingenen, R., and Raes, F.: Submicron aerosol mass balance at urban and semirural sites in the Milan area (Italy), J. Geophys. Res., 107, LOP 11–1–LOP 11–10, https://doi.org/10.1029/2000JD000111, 2002.

Putaud, J.-P., Dingenen, R. V., Alastuey, A., Bauer, H., Birmili, W., Cyrys, J., Flentje, H., Fuzzi, S., Gehrig, R., Hansson, H., Harrison, R., Herrmann, H., Hitzenberger, R., Hüglin, C., Jones, A., Kasper-Giebl, A., Kiss, G., Kousa, A., Kuhlbusch, T., Löschau, G., Maenhaut, W.,



Molnar, A., Moreno, T., Pekkanen, J., Perrino, C., Pitz, M., Puxbaum, H., Querol, X., Rodriguez, S., Salma, I., Schwarz, J., Smolik, J., Schneider, J., Spindler, G., ten Brink, H., Tursic, J., Viana, M., Wiedensohler, A., and Raes, F.: A European aerosol phenomenology – 3: Physical and chemical characteristics of particulate matter from 60 rural, urban, and kerbside sites across Europe, Atmos. Environ., 44, 1308 – 1320, https://doi.org/10.1016/j.atmosenv.2009.12.011, 2010.

Putaud, J. P., Cavalli, F., Martins dos Santos, S., and Dell'Acqua, A.: Long-term trends in aerosol optical characteristics in the Po Valley, Italy, Atmos. Chem. Phys., 14, 9129–9136, https://doi.org/10.5194/acp-14-9129-2014, 2014.

Ramanathan, V., Crutzen, P. J., Kiehl, J. T., and Rosenfeld, D.: Aerosols, Climate, and the Hydrological Cycle, Science, 294, 2119–2124, https://doi.org/10.1126/science.1064034, 2001.

Rosati, B., Gysel, M., Rubach, F., Mentel, T. F., Goger, B., Poulain, L., Schlag, P., Miettinen, P., Pajunoja, A., Virtanen, A., Klein Baltink,
H., Henzing, J. S. B., Größ, J., Gobbi, G. P., Wiedensohler, A., Kiendler-Scharr, A., Decesari, S., Facchini, M. C., Weingartner, E., and Baltensperger, U.: Vertical profiling of aerosol hygroscopic properties in the planetary boundary layer during the PEGASOS campaigns, Atmos. Chem. Phys., 16, 7295–7315, https://doi.org/10.5194/acp-16-7295-2016, 2016.

Saarikoski, S., Carbone, S., Decesari, S., Giulianelli, L., Angelini, F., Canagaratna, M., Ng, N. L., Trimborn, A., Facchini, M. C., Fuzzi, S., Hillamo, R., and Worsnop, D.: Chemical characterization of springtime submicrometer aerosol in Po Valley, Italy, Atmos. Chem. Phys.,
12, 8401–8421, https://doi.org/10.5194/acp-12-8401-2012, 2012.

Sabatier, T., Paci, A., Canut, G., Largeron, Y., Dabas, A., Donier, J.-M., and Douffet, T.: Wintertime Local Wind Dynamics from Scanning Doppler Lidar and Air Quality in the Arve River Valley, Atmosphere, 9, https://doi.org/10.3390/atmos9040118, 2018.

Samset, B. H.: How cleaner air changes the climate, Science, 360, 148–150, https://doi.org/10.1126/science.aat1723, 2018.

Sandrini, S., Fuzzi, S., Piazzalunga, A., Prati, P., Bonasoni, P., Cavalli, F., Bove, M. C., Calvello, M., Cappelletti, D., Colombi, C.,
Contini, D., de Gennaro, G., Di Gilio, A., Fermo, P., Ferrero, L., Gianelle, V., Giugliano, M., Ielpo, P., Lonati, G., Marinoni, A., Massabò, D., Molteni, U., Moroni, B., Pavese, G., Perrino, C., Perrone, M. G., Perrone, M. R., Putaud, J.-P., Sargolini, T., Vecchi, R., and Gilardoni, S.: Spatial and seasonal variability of carbonaceous aerosol across Italy, Atmos. Environ., 99, 587 – 598, https://doi.org/https://doi.org/10.1016/j.atmosenv.2014.10.032, 2014.

Schaap, M., van Loon, M., ten Brink, H. M., Dentener, F. J., and Builtjes, P. J. H.: Secondary inorganic aerosol simulations for Europe with
special attention to nitrate, Atmos. Chem. Phys., 4, 857–874, https://doi.org/10.5194/acp-4-857-2004, 2004.

Schmidli, J.: Daytime Heat Transfer Processes over Mountainous Terrain, J. Atmos. Sci., 70, 4041–4066, https://doi.org/10.1175/JAS-D-13-083.1, 2013.

Schmidli, J., Böing, S., and Fuhrer, O.: Accuracy of Simulated Diurnal Valley Winds in the Swiss Alps: Influence of Grid Resolution, Topography Filtering, and Land Surface Datasets, Atmosphere, https://doi.org/10.3390/atmos9050196, 2018.

Seibert, P., Kromp-Kolb, H., Baltensperger, U., Jost, D. T., and Schwikowski, M.: Trajectory Analysis of High-Alpine Air Pollution Data, pp. 595–596, Springer US, Boston, MA, https://doi.org/10.1007/978-1-4615-1817-4_65, 1994.

Seibert, P., Kromp-kolb, H., Kasper, A., Kalina, M., Puxbaum, H., Jost, D. T., Schwikowski, M., and Baltensperger, U.: Transport of polluted boundary layer air from the Po Valley to high-alpine sites, Atmos. Environ., 32, 3953 – 3965, https://doi.org/https://doi.org/10.1016/S1352-2310(97)00174-X, 1998.

Seinfeld, J. H. and Pandis, S. N.: Atmospheric Chemistry and Physics: From Air Pollution to Climate Change - 2nd ed., John Wiley & Sons, 2006.

Serafin, S. and Zardi, D.: Daytime Heat Transfer Processes Related to Slope Flows and Turbulent Convection in an Idealized Mountain Valley, J. Atmos. Sci., 67, 3739–3756, https://doi.org/10.1175/2010JAS3428.1, 2010.




Silibello, C., Calori, G., Brusasca, G., Giudici, A., Angelino, E., Fossati, G., Peroni, E., and Buganza, E.: Modelling of PM 10 concentrations over Milano urban area using two aerosol modules, Environ. Modell. Softw., 23, 333–343, https://doi.org/10.1016/j.envsoft.2007.04.002, 2008.

Sprenger, M. and Wernli, H.: The LAGRANTO Lagrangian analysis tool – version 2.0, Geosci. Model Dev., 8, 2569–2586, https://doi.org/10.5194/gmd-8-2569-2015, 2015.

Squizzato, S. and Masiol, M.: Application of meteorology-based methods to determine local and external contributions to particulate matter pollution: A case study in Venice (Italy), Atmos. Environ., 119, 69 – 81, https://doi.org/https://doi.org/10.1016/j.atmosenv.2015.08.026, 2015.

Stohl, A.: Trajectory statistics-A new method to establish source-receptor relationships of air pollutants and its application to the transport of particulate sulfate in Europe, Atmos. Environ., 30, 579 – 587, https://doi.org/https://doi.org/10.1016/1352-2310(95)00314-2, 1996.

Tampieri, F., Trombetti, F., and Scarani, C.: Summer daily circulation in the Po Valley, Italy, Geophys. Astro. Fluid, 17, 97–112, https://doi.org/10.1080/03091928108243675, 1981.

Tang, L., Haeger-Eugensson, M., Sjöberg, K., Wichmann, J., Molnár, P., and Sallsten, G.: Estimation of the long-range transport contribution from secondary inorganic components to urban background PM10 concentrations in south-western Sweden during 1986–2010, Atmos. Environ., 89, 93 – 101, https://doi.org/https://doi.org/10.1016/j.atmosenv.2014.02.018, 2014.

Teixeira, M. A. C., Kirshbaum, D. J., Ólafsson, H., Sheridan, P. F., and Stiperski, I., eds.: The Atmosphere over Mountainous Regions, Frontiers Research Topics, 2016.

Thunis, P., Pederzoli, A., and D., P.: Performance criteria to evaluate air quality modeling applications, Atmos. Environ., 59, 476 – 482, https://doi.org/https://doi.org/10.1016/j.atmosenv.2012.05.043, 2012.

Thyer, N. H.: A theoretical explanation of mountain and valley winds by a numerical method, Archiv für Meteorologie, Geophysik und Bioklimatologie, Serie A, 15, 318–348, https://doi.org/10.1007/BF02247220, 1966.

Tudoroiu, M., Eccel, E., Gioli, B., Gianelle, D., Schume, H., Genesio, L., and Miglietta, F.: Negative elevation-dependent warming trend in the Eastern Alps, Environ. Res. Lett., 11, 044 021, https://doi.org/10.1088/1748-9326/11/4/044021, 2016.

Van Donkelaar, A., Martin, R. V., Brauer, M., Kahn, R., Levy, R., Verduzco, C., and Villeneuve, P. J.: Global estimates of ambient fine particulate matter concentrations from satellite-based aerosol optical depth: development and application, Environ. Health Persp., 118, 847, https://doi.org/10.1289/ehp.0901623, 2010.

Wagner, J. S., Gohm, A., and Rotach, M. W.: The impact of valley geometry on daytime thermally driven flows and vertical transport processes, Q. J. Roy. Meteor. Soc., 141, 1780–1794, https://doi.org/10.1002/qj.2481, 2014.

Waked, A., Favez, O., Alleman, L. Y., Piot, C., Petit, J.-E., Delaunay, T., Verlinden, E., Golly, B., Besombes, J.-L., Jaffrezo, J.-L., and Leoz-Garziandia, E.: Source apportionment of PM$_{10}$ in a north-western Europe regional urban background site (Lens, France) using positive matrix factorization and including primary biogenic emissions, Atmos. Chem. Phys., 14, 3325–3346, https://doi.org/10.5194/acp-14-3325-2014, 2014.

Wang, X., Wang, W., Yang, L., Gao, X., Nie, W., Yu, Y., Xu, P., Zhou, Y., and Wang, Z.: The secondary formation of inorganic aerosols in the droplet mode through heterogeneous aqueous reactions under haze conditions, Atmos. Environ., 63, 68 – 76, https://doi.org/https://doi.org/10.1016/j.atmosenv.2012.09.029, 2012.

Weissmann, M., Braun, F. J., Gantner, L., Mayr, G. J., Rahm, S., and Reitebuch, O.: The Alpine Mountain–Plain Circulation: Airborne Doppler Lidar Measurements and Numerical Simulations, Mon. Weather Rev., 133, 3095–3109, https://doi.org/10.1175/MWR3012.1, 2005.





WHO: Ambient air pollution: a global assessment of exposure and burden of disease, Tech. rep., 2016.

Wiegner, M. and Geiß, A.: Aerosol profiling with the Jenoptik ceilometer CHM15kx, Atmos. Meas. Tech., 5, 1953–1964, https://doi.org/10.5194/amt-5-1953-2012, 2012.

WMO: WMO/IGAC Impacts of megacities on air pollution and climate, Tech. rep., World Meteorological Organization, https://library.wmo.int/pmb_ged/gaw_205.pdf, 2012.

Wotawa, G., Kröger, H., and Stohl, A.: Transport of ozone towards the Alps – results from trajectory analyses and photochemical model studies, Atmos. Environ., 34, 1367 – 1377, https://doi.org/https://doi.org/10.1016/S1352-2310(99)00363-5, 2000.

Ying, C., Chunsheng, Z., Qiang, Z., Zhaoze, D., Mengyu, H., and Xincheng, M.: Aircraft study of Mountain Chimney Effect of Beijing, China, J. Geophys. Res., 114, https://doi.org/10.1029/2008JD010610, 2009.

Zardi, D. and Whiteman, C. D.: Diurnal Mountain Wind Systems, pp. 35–119, Springer Netherlands, Dordrecht, https://doi.org/10.1007/978-94-007-4098-3_2, 2013.

Zeng, Y. and Hopke, P.: A study of the sources of acid precipitation in Ontario, Canada, Atmos. Environ., 23, 1499 – 1509, https://doi.org/https://doi.org/10.1016/0004-6981(89)90409-5, 1989.

Zeng, Z., Chen, A., Ciais, P., Li, Y., Li, L. Z. X., Vautard, R., Zhou, L., Yang, H., Huang, M., and Piao, S.: Regional air pollution brightening reverses the greenhouse gases induced warming-elevation relationship, Geophys. Res. Lett., 42, 4563–4572, https://doi.org/10.1002/2015GL064410, 2015.

Zong, Z., Wang, X., Tian, C., Chen, Y., Fu, S., Qu, L., Ji, L., Li, J., and Zhang, G.: PMF and PSCF based source apportionment of PM2.5 at a regional background site in North China, Atmos. Res., 203, 207 – 215, https://doi.org/https://doi.org/10.1016/j.atmosres.2017.12.013, 2018.

Zuev, V. V., Burlakov, V. D., Nevzorov, A. V., Pravdin, V. L., Savelieva, E. S., and Gerasimov, V. V.: 30-year lidar observations of the stratospheric aerosol layer state over Tomsk (Western Siberia, Russia), Atmos. Chem. Phys., 17, 3067–3081, https://doi.org/10.5194/acp-17-3067-2017, 2017.