# Peer review of "Transport of Po Valley aerosol pollution to the northwestern Alps – Part 2: Long-term impact on air quality"

_Atmospheric Chemistry and Physics, 2019_

## Referee Comment (RC1) · Anonymous Referee #1 · 14 Apr 2019

General Comments The paper evaluates the impact of trans-regional transport of aerosol from the Po Plain to the Aosta Valley (north-western Italian Alps), by means of both the analysis of experimental data and numerical simulations. The paper is complete, well-written and can be of interest for the community. Therefore in my opinion it is worth publishing in Atmospheric Chemistry and Physics, after a few minor comments are addressed.

Specific Comments 1) Meteorological model: the Authors say (pag. 8, line 7) that "we used a nudged, high-resolution variant, called COSMO-I2, covering Italy". However, in my opinion 2.8 km is not high-resolution in complex terrain. The floor of the Aosta

[Figure]

Valley is 2-3 km wide, so meteorological phenomena triggered by the orography may be not well resolved by the model at this resolution, also considering the fact that the actual model resolution is 6-8 times the grid cell (Skamarock 2004). In fact authors say that the model surface altitude of Aosta urban area is 900 m a.s.l., whereas the actual altitude is 600 m a.s.l. For example Schmidli et al. (2018) showed that at 2.2-km resolution the COSMO model poorly simulates valley winds, while at 1.1-km resolution the diurnal cycle of the valley winds is well represented. Similarly, Giovannini et al. (2014) showed that 2-km resolution can be considered as the limit for a good representation of valley winds in narrow Alpine valleys. 2) Building on the previous consideration, it is not easy to evaluate the performance of the chemical model without a validation of the meteorological model. The Authors attribute most of the discrepancies with respect to measurements to deficiencies in the boundary conditions of the chemical model. However, this statement is difficult to be demonstrated without a complete validation of the modelling chain. Moreover, increasing the boundary conditions by a factor 4 reduces the mean bias, but does not reduce the RMSE (Table 3). So it seems that there is still a compensation of underestimations and overestimations. 3) In Figure 17 it would be interesting to see also a normalized mean bias (i.e. a mean bias normalized by the average concentration measured for each class).

Technical remarks Page 1, line 4: ...3-year period... Section 4.3: The two final sentences begin with "Finally". Page 36, line 18: ...this regular air mass transport...

References Giovannini, L.; Antonacci, G.; Zardi, D.; Laiti, L.; Panziera, L. Sensitivity of simulated wind speed to spatial resolution over complex terrain. Energy Procedia, 2014, 59, 323-329. Schmidli, J.; Böing, S.; Fuhrer, O. Accuracy of simulated diurnal valley winds in the Swiss Alps: influence of grid resolution, topography filtering, and land surface datasets. Atmosphere, 2018, 9, 196. Skamarock, W.C. Evaluating mesoscale NWP models using kinetic energy spectra. Mon. Weather Rev., 2004, 132, 3019-3032.

---

## Referee Comment (RC2) · Anonymous Referee #2 · 15 Apr 2019

Manuscript acp-2019-128
**Transport of Po Valley aerosol pollution to the northwestern Alps.**
**Part 2: Long-term impact on air quality**

Introduction

The manuscript presents a comprehensive analysis of meteorological (standard weather parameters, ceilometer) and air quality (PM10 characterization) measurements over a three-year period in the Aosta Valley in north-western Italy. The study convincingly demonstrates that a non-negligible fraction of the particulate matter concentration observed at receptor sites in the valley is advected from the neighboring Po plain, occasionally providing a decisive contribution to the exceedance of concentration limit values. The study also shows that advected aerosols differ from the locally emitted ones in their physical-chemical properties. Results from a chemical transport modelling chain are evaluated, demonstrating that the effects of advection from the Po plain can be represented only qualitatively from state-of-the art operational model chains.

Recommendation

Methodology and results are scientifically sound. The manuscript is written in good English and figures are generally of adequate quality. Suggestions for a few minor modifications should be taken into account before publication can be recommended. Details are provided below. Please note that the assessment is based mostly on the meteorological aspects of the study, reflecting the expertise of the reviewer.

Comments

1. The superficial reader might conclude that pollution in the Aosta valley mostly depends on advection from the Po plain. While this is true to some extent, one should also consider that easterly advection is most common in summer (Figure 3), that is, when PM10 concentrations in Aosta are at their yearly minimum (Figure 8). Based on these contrasting statements, another superficial reader (with an opposite bias) would conclude that easterly advection actually cleans the air. The latter is obviously a flawed argument, still it shows that it is important to carefully delimit the message of the study.
A balanced perspective on the whole issue could be the following: easterly advection from the Po valley, which is most frequent in summer but possible in all seasons, may combine with adverse weather conditions (low-level inversions) locally worsened by the valley topography, greatly enhancing pollution levels. This is especially true in the coldest part of the year, when emissions are highest.
I feel that, in the introductory and concluding remarks, the authors could do a better job in explaining these subtleties.
2. The abstract is very long. ACP doesn't seem to set length limits, but it is in the interest of the authors to provide a more concise formulation. Please refrain from using references to published literature in the abstract.
3. Page 1, lines 17-18. "maximises" → "is highest", "minimizes" → "is lowest".
4. Page 2, line 27. There is a double blank space before "WMO".
5. Page 3, Figure 1. Please align image borders (merely for aesthetic reasons).
6. Page 3, line 15. Remove "the main of which are".
7. Page 4, line 31. Thermally-driven winds include a night-time component as well (drainage flows along slopes and downvalley winds), which can ventilate the urban atmosphere and reduce pollutant loads. Are nocturnal breezes not observed in Aosta?

8. Page 4, line 32. Strictly speaking, foehn is not necessarily a rain-shadow wind. Even if the (quite inaccurate) textbook picture of Foehn emphasizes the impact of upstream latent heat release due to condensation, Foehn is in most cases connected only to low-level blocked flow upstream of the orographic obstacle. Foehn may develop even if there is no upstream precipitation. Warming mostly occurs through the adiabatic descent of unblocked air from levels above the mountain tops (see for instance this DOI: 10.1127/0941-2948/2012/0398).

9. Page 7, table 1. Footnotes b and c. Unclear: is this the average data availabilty, or are data regularly available with this periodicity?

10. Page 8, line 7. I am not sure that COSMO-I2 is a "high-resolution" model. Its grid spacing is certainly in line with that of other state-of-the-art operational limited-area-models. However, it is barely sufficient to appropriately resolve thermally-driven circulations. See for instance this DOI: 10.1175/MWR-D-14-00002.1. Please consider also comment 21 below.

11. Page 9, lines 26-30. How sensitive are the TSM model results to the ad-hoc method of weighting the concentration fields on the number of trajectories through each cell? How si the weighting performed, exactly? Does it only reduce noise in the spatial fields, as stated, or does it also affect their magnitude?

12. Page 10, line 20. "...was chosen based on the Q/Qexp ratio". Please clarify.

13. Page 13, table 2. It seems to me that the criteria outlined here are not mutually exclusive. For instance, would a 13-hour period with wind speed exceeding 4 m/s be classified as "downwind" or as "foehn"? I also have a few concerns about the terminology.
(A) "Upwind" and "downwind" typically identify position relative to the wind and to a specific location (e.g. "when the flow is southerly, the city is downwind of the mountains"). In this context, it might be more appropriate to speak of "Channeled synoptic flow from the west/east". In addition, it would be appropriate to specify wind direction ranges for the two classes.
(B) "Breezes" have by definition wind speed > 1.5 m/s (Beaufort scale). This would exclude wind calms at night . It might be better to speak of "Diurnal wind system".
(C) "Stability" refers to the thermal structure of the atmosphere. Because winds are emphasized here, it might be appropriate to speak of "Wind calm".
(D) The directional range for the Foehn class probably deserves being explained.
(E) Please use wind "direction" instead of "provenance". "Direction" refers conventionally to the direction the wind comes from.

14. Page 14, figure 4. In summer, about 70% of the days feature breeze systems, but aerosol-layer days are only about 50% (Figure 3). What about the rest? Are easterly breezes occasionally "clean"?

15. Page 15, line 14. It would be useful to explain where the Divedro Valley is. The great majority of readers wouldn't have a clue otherwise.

16. Page 16, line 15. Please remove the comma between "simple" and "forecasting".

17. Page 19, line 4. Reference to Teixeira et al 2016. The interested reader wouldn't find much in this editorial. More appropriate references could be those available at these DOIs: 10.3389/feart.2015.00077 and https://doi.org/10.3390/atmos9030102.

18. Page 25, line 5. The dashed continuous line in Figure 12a is very hard to see. Consider replacing with a colored line, or with a filled area in the background.

19. Page 25, Figure 13. Text is hard to read on the colored background. Consider using a different color map, or adding white background to the location labels.

20. Pages 30-32. There is some redundancy in Figures 16-17-18-19. It is unclear why results from two locations are represented in the figure. In fact, the differences between the two sites are never discussed in detail in the text. It would be possible to remove the figure panels referring to Donnas without great loss of information.

21. Page 33, lines 2-3. "This is a clear indication that the external contribution (boundary conditions) is not optimally parametrised in the model". I am really not sure that I agree on this statement. If I understand correctly, FARM uses information from the whole COSMO-I2 domain, which completely includes the Po valley. Therefore, the problem does not lie in "external contributions" not being represented in the lateral boundary conditions. There might be problems with the lower boundary conditions (emission inventories), but I doubt these can be wrong by a factor of 4 over the whole Po valley. I am not even sure that model parameterisations matter here.

    I would argue that the relatively coarse resolution of the weather model (2.8 km, marginally enough to resolve the valley) and of the transport model (only 16 vertical levels) play an important role. Please consider the topography cross-sections in Figure 20. Aosta seems to lie in a basin, likely non-existent in reality and introduced in the model by terrain smoothing. If the topography profile descended continuously from Aosta to Donnas and the Po Plain (as in reality), it would be possible to resolve the horizontal advection of aerosol-laden air from much lower altitudes on the plain, likely removing a large fraction of the negative concentration bias.

22. Page 33, line 10. "The reason for discrepancies is therefore likely in the emissions". Why not in deficiencies of the transport modelling, as I argued above?

23. Page 46, line 18: There seems to be a problem with the reference to Thunis et al, 2012 (the third author).

---

## Author Comment (AC1) · 5 Jun 2019

**Response to Anonymous Referee #1**

General Comment. The paper evaluates the impact of trans-regional transport of aerosol from the Po Plain to the Aosta Valley (north-western Italian Alps), by means of both the analysis of experimental data and numerical simulations. The paper is complete, well-written and can be of interest for the community. Therefore in my opinion it is worth publishing in Atmospheric Chemistry and Physics, after a few minor comments are addressed.

We thank the reviewer for taking the time to revise our manuscript and for his/her pertinent comments. Our point-to-point reply is given hereafter (the text in italics represents a citation of the revised manuscript and the figure references follow the updated numbering).

Referee's comment 1. Meteorological model: the Authors say (pag. 8, line 7) that "we used a nudged, high-resolution variant, called COSMO-I2, covering Italy". However, in my opinion 2.8 km is not high-resolution in complex terrain. The floor of the Aosta Valley is 2-3 km wide, so meteorological phenomena triggered by the orography may be not well resolved by the model at this resolution, also considering the fact that the actual model resolution is 6-8 times the grid cell (Skamarock 2004). In fact authors say that the model surface altitude of Aosta urban area is 900 m a.s.l., whereas the actual altitude is 600 m a.s.l. For example Schmidli et al. (2018) showed that at 2.2-km resolution the COSMO model poorly simulates valley winds, while at 1.1-km resolution the diurnal cycle of the valley winds is well represented. Similarly, Giovannini et al. (2014) showed that 2-km resolution can be considered as the limit for a good representation of valley winds in narrow Alpine valleys.

Author's response 1. We understand the point raised by the reviewer. In the submitted version, we used the terminology "high resolution", as this appears in the COSMO web page (http://www.cosmo-model.org/content/tasks/operational/default.htm), where COSMO is indeed defined as a "high-resolution" model.

However, we accept the reviewer's remark and thus removed the term "high-resolution" in the revised text. Also, in Sect. 3.1 we now discuss the advantages of COSMO-I2 version over COSMO-ME, with reference to the high*er* resolution of the former (but not "high" in absolute terms). The revised text reads as follows:

The forecasts, inclusive of the complete set of parameters (such as the 3-D wind velocity used here) for eight time steps (from 00 to 21 UTC), are disseminated daily in two different configurations by the meteo-rological operative centre – air force meteorological service (COMET): a lower-resolution version (COSMO-ME, 7 km horizontal grid and 45 levels vertical grid, 72 hours integration), covering central and southern

Europe, and a nudged, higher-resolution version (COSMO-I2 or COSMO-IT, 2.8 km, 65 vertical levels, 2 runs/day), covering Italy (orange rectangle in Fig. 1a). Owing to the complex topography of the Aosta Valley, and the consequent need to resolve as much as possible the atmospheric circulation at small spatial scales, we used the latter version in the present work (cf. Sect. 4.5 for a discussion about possible effects of the finite model resolution in complex terrain).

Furthermore, following the reviewer's comment, we now preferably use the term "grid spacing/step" instead of "resolution" throughout the revised text.

We also added the suggested references in Sect. 4.5:

Although the COSMO-I2 grid spacing is certainly in line with that of other state-of-the-art operational limited-area models, in our complex terrain it could be insufficient to appropriately resolve local meteorological phenomena triggered by the valley orography (Wagner et al., 2014), also considering that the actual model resolution is 6–8 times the grid cell (Skamarock, 2004). For example, Schmidli et al. (2018) show that at 2.2-km grid step the COSMO model poorly simulates valley winds, while at 1.1-km grid step the diurnal cycle of the valley winds is well represented. Similarly, Giovannini et al. (2014) show that 2km step can be considered as the limit for a good representation of valley winds in narrow Alpine valleys. Smoothed digital elevation model (DEM) used within COSMO and FARM could also play a direct role in the detected underestimation. In fact, as mentioned in Sect.3.2, the model surface altitude of the Aosta urban area is 900 m a.s.l., whereas the actual altitude is about 580 m a.s.l. (Table 1). The adjacent cells are given an even higher altitude, owing to the fact that the valley floor and the neighbouring mountain slopes are not properly resolved at the current resolution. This results in an apparent lower elevation of the sites located in flatter and wider areas, such as Aosta, while the real topography profile at the bottom of the valley presents a monotonic increase from the Po plain to Aosta. Therefore, it is expected that, just by better reproducing the orography, a higher resolution would allow to better resolve the horizontal advection of aerosol-laden air from much lower altitudes on the plain.

Most of these issues were extensively addressed in the companion paper (Diémoz et al., 2019). In that study, COSMO-I2 was shown to be capable of reproducing the mountain-plain wind patterns observed at the surface both on average (cf. Fig. S1 in the companion paper) and in specific case studies (Fig. S13 therein). Nevertheless, it was also found to slightly anticipate in time and overestimate the easterly diurnal winds in the first hours of the afternoon and to overestimate the nighttime drainage winds (katabatic winds, ventilating the urban atmosphere and reducing pollutant loads). These limitations were mostly attributed to the finite resolution of the model.

RC2. Building on the previous consideration, it is not easy to evaluate the performance of the chemical model without a validation of the meteorological model. The Authors attribute most of the discrepancies with respect to measurements to deficiencies in the boundary conditions of the chemical model. However, this statement is difficult to be demonstrated without a complete validation of the modelling chain. Moreover, increasing the boundary conditions by a factor 4 reduces the mean bias, but does not reduce the RMSE (Table 3). So it seems that there is still a compensation of underestimations and overestimations.

AR2. One of the most relevant updates in the revised manuscript is the study of the chemical transport model performances in a flatter terrain compared to the complex orography of the mountain valley. This is accomplished by including the  $PM_{10}$  dataset measured in a station at the boundary of the domain, Ivrea. This addition clearly showed that the same  $PM_{10}$  underestimation of the model is also obtained over flat terrain, demonstrating this is not driven by inaccuracies of the simulated wind fields over the more complex terrain of Aosta. Therefore, Sect. 4.5 has been modified to better explain this point and investigate the relative importance of both emissions (inventories/unaccounted physical processes) and transport (performances of the meteorological model) in generating the model-measurements differences:

[revised manuscript text omitted]

We report the new figures 16 and 19, previously mentioned in the text, here below. Fig. 17 is shown in AR3.

**Figure 16:** Long-term (1 year) comparison between PM10 surface measurements and simulations (FARM) in Aosta–Downtown (a) and Ivrea (b).

---

## Author Comment (AC2) · 5 Jun 2019

**Response to Anonymous Referee #2**

**General Comment. The manuscript presents a comprehensive analysis of meteorological (standard weather parameters, ceilometer) and air quality (PM10 characterization) measurements over a three-year period in the Aosta Valley in north-western Italy. The study convincingly demonstrates that a non-negligible fraction of the particulate matter concentration observed at receptor sites in the valley is advected from the neighboring Po plain, occasionally providing a decisive contribution to the exceedance of concentration limit values. The study also shows that advected aerosols differ from the locally emitted ones in their physical-chemical properties. Results from a chemical transport modelling chain are evaluated, demonstrating that the effects of advection from the Po plain can be represented only qualitatively from state-of-the art operational model chains.**

**Methodology and results are scientifically sound. The manuscript is written in good English and figures are generally of adequate quality. Suggestions for a few minor modifications should be taken into account before publication can be recommended. Details are provided below. Please note that the assessment is based mostly on the meteorological aspects of the study, reflecting the expertise of the reviewer.**

We thank the reviewer for taking the time to revise our manuscript and for his/her pertinent comments. Our point-to-point reply is given hereafter (the text in italics represents a citation of the revised manuscript and the figure references follow the updated numbering).

**Referee's comment 1. The superficial reader might conclude that pollution in the Aosta valley mostly depends on advection from the Po plain. While this is true to some extent, one should also consider that easterly advection is most common in summer (Figure 3), that is, when PM10 concentrations in Aosta are at their yearly minimum (Figure 8). Based on these contrasting statements, another superficial reader (with an opposite bias) would conclude that easterly advection actually cleans the air. The latter is obviously a flawed argument, still it shows that it is important to carefully delimit the message of the study. A balanced perspective on the whole issue could be the following: easterly advection from the Po valley, which is most frequent in summer but possible in all seasons, may combine with adverse weather conditions (low-level inversions) locally worsened by the valley topography, greatly enhancing pollution levels. This is especially true in the coldest part of the year, when emissions are highest. I feel that, in the introductory and concluding remarks, the authors could do a better job in explaining these subtleties.**

Author's response 1. We thank the reviewer for raising this possible ambiguity. In the revised manuscript (Sect. 4.3.2), we now split the impact of transport on $PM_{10}$ levels by season. In this way, we better show

that the overall impact actually depends on the coupling between frequency and severity of the episodes, and also on the background local pollution levels. The revised text reads as follows:

*On a seasonal basis, the absolute impact of air masses transport depends on the coupling between emissions (stronger in winter and weaker in summer) and weather regimes (e.g., thermal winds occurring more frequently in summer/autumn with respect to winter/spring, Sect. 4.1). This results in a $PM_{10}$ contribution of nearly 6 $\mu g\ m^{-3}$ in winter and autumn, 4 $\mu g\ m^{-3}$ in summer, and 3 $\mu g\ m^{-3}$ in spring. In terms of relative contribution, this also depends on the "background" $PM_{10}$ levels in Aosta. It is therefore highest in summer (32%), when thermally-driven fluxes are more frequent and local emissions lower, and lowest in winter (16%), when thermal winds are less frequent and local emissions higher. Intermediate relative values are found in spring and summer (27% and 28%, respectively). In specific episodes,* advections from the Po basin may still produce an increase of (...)*

Following the reviewer's comment, we also updated the Conclusions accordingly:

*Using their relevant scores as a proxy, we estimate an average 25% contribution of these transported nitrate and sulfate-rich components to the Aosta $PM_{10}$. This impact varies on a seasonal basis. The relative contribution of non-local $PM_{10}$ is highest in summer (32%), when advection is most frequent and local $PM_{10}$ is lowest, while it is lowest in winter (16%), when advection is least frequent and local $PM_{10}$ is highest. In absolute terms the reverse occurs and the impact of transport is found to be highest in winter/autumn, reaching levels of 50 $\mu g\ m^{-3}$ in some episodes (thus exceeding alone the EU legislative limits). This occurs due to the superposition of advected particles with higher background concentrations in the coldest part of the year, when emissions are highest and pollution is further enhanced by adverse weather conditions, i.e. low-level inversions, locally worsened by the valley topography. Note, though, that in this study the impact of Po Valley advection on air quality was mostly quantified for the Aosta–Downtown station. In rural and remote sites of the region this non-local contribution is expected to be even larger, in relative terms (...)*

Finally, where we now specify already in the Introduction that the impact on air quality depends *on the interplay between frequency and severity of the episodes.*

**RC2. The abstract is very long. ACP doesn't seem to set length limits, but it is in the interest of the authors to provide a more concise formulation. Please refrain from using references to published literature in the abstract.**

AR2. Thank you for this suggestion. We agree with it. The abstract has been shortened by more than 700 characters and bibliographic references have been removed. It now reads:

*This work evaluates the impact of trans-regional aerosol transport from the polluted Po basin on particulate matter levels ($PM_{10}$) and physico-chemical characteristics in the northwestern Alps. To this purpose, we exploited a multi-sensor, multi-platform database over a 3-year period (2015–2017) accompanied by a series of numerical simulations. The experimental setup included operational (24/7) vertically resolved aerosol profiles by an Automated LiDAR-Ceilometer (ALC), vertically integrated aerosol properties by a sun/sky photometer, and surface measurements of aerosol mass concentration, size distribution and chemical composition. This experimental set of observations was then complemented by modelling tools, including Numerical Weather Prediction (NWP), Trajectory Statistical (TSM) and Chemical Transport (CTM) models, plus Positive Matrix Factorisation (PMF) on both the $PM_{10}$ chemical speciation analyses and particle size distributions. In a first companion study, we showed and discussed through detailed case studies the 4-D phenomenology of recurrent episodes of aerosol transport from the polluted Po basin to the northwestern Italian Alps. Here we draw more general and statistically significant conclusions on the frequency of occurrence of this phenomenon, and on the quantitative impact of this regular, wind-driven, aerosol-rich "atmospheric tide" on $PM_{10}$ air quality levels in this alpine environment. Based on an original ALC-derived classification, we found that an advected aerosol layer is observed at the receptor site*

*(Aosta) in 93% of days characterized by easterly winds (i.e., from the Po basin) and that the longer the time spent by air masses over the Po plain the higher this probability. Frequency of these advected aerosol layers was found to be rather stable over the seasons with about 50% of the days affected. Duration of these advection events ranges from few hours up to several days, while aerosol layer thickness ranges from 500 up to 4000 m. Our results confirm this phenomenon to be related to non-local emissions, to act at the regional scale and to largely impact both surface levels and column-integrated aerosol properties. In Aosta, $PM_{10}$ and AOD values increase respectively up to a factor of 3.5 and 4 in dates under the Po Valley influence. Pollution transport events were also shown to modify the mean chemical composition and typical size of particles in the target region. In fact, increase in secondary species, and mainly nitrate- and sulfate-rich components, were found to be effective proxies of the advections, with the transported aerosol responsible for at least 25% of the $PM_{10}$ measured in the urban site of Aosta, and adding up to over 50 $\mu g\ m^{-3}$ during specific episodes (thus exceeding alone the EU established daily limit). From a modelling point of view, our CTM simulations performed over a full year showed that the model is able to reproduce the phenomenon, but markedly underestimates its impact on $PM_{10}$ levels. As a sensitivity test, we employed the ALC-derived identification of aerosol advections to re-weight the emissions from outside the boundaries of the regional domain in order to match the observed $PM_{10}$ field. This simplified exercise indicated that an increase of such "external" emissions by a factor of 4 in the model is needed to halve the model $PM_{10}$ maximum deviations and to significantly reduce the $PM_{10}$ normalised mean bias forecasts error (from -35% to 5%).*

**RC3. Page 1, lines 17-18. "maximises" → "is highest", "minimizes" → "is lowest".**

AR3. The wording has been changed in the new abstract (see AR2).

**RC4. Page 2, line 27. There is a double blank space before "WMO".**

AR4. Corrected.

**RC5. Page 3, Figure 1. Please align image borders (merely for aesthetic reasons).**

AR5. Done. The updated Fig. 1 is shown here below. This also includes other modifications as required by the revision process.

[Figure]

[Figure]

**Figure 1:** True colour corrected reflectance from MODIS Terra satellite (`http://worldview.earthdata.nasa.gov`) on 17 March 2017. (a) Italy, with indication of the Alpine and Po Valley regions, of the Aosta Valley FARM regional domain (light blue rectangle), and the COSMO-I2 domain (orange rectangle, approximately corresponding to the boundaries of the national inventory). (b) Zoom over the Aosta Valley. The circle markers represent the sites of 1) Aosta–Downtown; 2) Aosta–Saint-Christophe; 3) Donnas; 4) Ivrea. A thin aerosol layer over the Po basin starting to spread out into the Alpine valleys is visible in both figures.

**RC6. Page 3, line 15. Remove "the main of which are".**

AR6. Removed.

**RC7. Page 4, line 31. Thermally-driven winds include a night-time component as well (drainage flows along slopes and downvalley winds), which can ventilate the urban atmosphere and reduce pollutant loads. Are nocturnal breezes not observed in Aosta?**

AR7. Thanks for pointing this out. Indeed, we extensively studied nocturnal breezes in the companion paper (Diémoz et al., 2019), since they can be relevant for the daily cycle of pollutant concentrations. Some of those results are now mentioned in the revised manuscript (Sect. 4.5):

*(...) Most of these issues were extensively addressed in the companion paper (Diémoz et al., 2019). In that study, COSMO-I2 was shown to be capable of reproducing the mountain-plain wind patterns observed at the surface both on average (cf. Fig. S1 in the companion paper) and in specific case studies (Fig. S13 therein). Nevertheless, it was also found to slightly anticipate in time and overestimate the easterly diurnal winds in the first hours of the afternoon and to overestimate the nighttime drainage winds (katabatic winds, ventilating the urban atmosphere and reducing pollutant loads). These limitations were mostly attributed to the finite resolution of the model.*

We have also updated the Introduction, which now reads:

The topography of the area triggers some of the most common weather regimes in the mountains, such as thermally-driven, *up-valley (daytime) and down-valley (nighttime) winds, up-slope (daytime) and*

*down-slope (nighttime) winds, "Foehn" winds (Seibert, 2012)* from the west, and frequent temperature inversions during wintertime anticyclonic days (...)

**RC8. Page 4, line 32. Strictly speaking, Foehn is not necessarily a rain-shadow wind. Even if the (quite inaccurate) textbook picture of Foehn emphasizes the impact of upstream latent heat release due to condensation, Foehn is in most cases connected only to low-level blocked flow upstream of the orographic obstacle. Foehn may develop even if there is no upstream precipitation. Warming mostly occurs through the adiabatic descent of unblocked air from levels above the mountain tops (see for instance this DOI: 10.1127/0941-2948/2012/0398).**

AR8. Thank you for this remark. We have removed the term "rain-shadow" from the text and we have included the suggested reference (the revised paragraph is already reported in AR7).

**RC9. Page 7, table 1. Footnotes b and c. Unclear: is this the average data availabilty, or are data regularly available with this periodicity?**

AR9. A short sentence has been added to Sect. 2 to better describe the *laboratory schedule (i.e., metal analyses on two consecutive days, EC/OC on the following two days, followed by two sequences metal-metal-EC/OC)*. Therefore, *over a 10 days period EC/OC concentrations are routinely provided on 4 days on average and metal concentrations on 6 days.*

The two footnotes of Table 1 now read: *The analysis is performed on 4* (or 6) *out of 10 days according to the laboratory schedule.*

**RC10. Page 8, line 7. I am not sure that COSMO-I2 is a "high-resolution" model. Its grid spacing is certainly in line with that of other state-of-the-art operational limited-area-models. However, it is barely sufficient to appropriately resolve thermally-driven circulations. See for instance this DOI: 10.1175/MWR-D-14-00002.1. Please consider also comment 21 below.**

AR10. We understand the point raised by the reviewer. In the submitted version, we used the terminology "high resolution", as this appears in the COSMO web page (`http://www.cosmo-model.org/content/tasks/operational/default.htm`), where COSMO is indeed defined as a "high-resolution" model.

However, we accept the reviewer's remark and thus removed the term "high-resolution" in the revised text. Also, in Sect. 3.1 we now discuss the advantages of COSMO-I2 version over COSMO-ME, with reference to the high*er* resolution of the former (but not "high" in absolute terms). The revised text reads as follows:

*The forecasts, inclusive of the complete set of parameters (such as the 3-D wind velocity used here) for eight time steps (from 00 to 21 UTC), are disseminated daily in two different configurations by the meteorological operative centre – air force meteorological service (COMET): a lower-resolution version (COSMO-ME, 7 km horizontal grid and 45 levels vertical grid, 72 hours integration), covering central and southern Europe, and a nudged, higher-resolution version (COSMO-I2 or COSMO-IT, 2.8 km, 65 vertical levels, 2*

*runs/day), covering Italy (orange rectangle in Fig. 1a). Owing to the complex topography of the Aosta Valley, and the consequent need to resolve as much as possible the atmospheric circulation at small spatial scales, we used the latter version in the present work (cf. Sect. 4.5 for a discussion about possible effects of the finite model resolution in complex terrain).*

Furthermore, following the reviewer's comment, we now preferably use the term "grid spacing/step" instead of "resolution" throughout the revised text.

We have also included some of the reviewer's comments in Sect. 4.5:

*Although the COSMO-I2 grid spacing is certainly in line with that of other state-of-the-art operational limited-area models, in our complex terrain it could be insufficient to appropriately resolve local meteorological phenomena triggered by the valley orography (Wagner et al., 2014), also considering that the actual model resolution is 6–8 times the grid cell (Skamarock, 2004). For example, Schmidli et al. (2018) show that at 2.2-km grid step the COSMO model poorly simulates valley winds, while at 1.1-km grid step the diurnal cycle of the valley winds is well represented. Similarly, Giovannini et al. (2014) show that 2-km step can be considered as the limit for a good representation of valley winds in narrow Alpine valleys. Smoothed digital elevation model (DEM) used within COSMO and FARM could also play a direct role in the detected underestimation. In fact, as mentioned in Sect.3.2, the model surface altitude of the Aosta urban area is 900 m a.s.l., whereas the actual altitude is about 580 m a.s.l. (Table 1). The adjacent cells are given an even higher altitude, owing to the fact that the valley floor and the neighbouring mountain slopes are not properly resolved at the current resolution. This results in an apparent lower elevation of the sites located in flatter and wider areas, such as Aosta, while the real topography profile at the bottom of the valley presents a monotonic increase from the Po plain to Aosta. Therefore, it is expected that, just by better reproducing the orography, a higher resolution would allow to better resolve the horizontal advection of aerosol-laden air from much lower altitudes on the plain.*

**RC11. Page 9, lines 26-30. How sensitive are the TSM model results to the ad-hoc method of weighting the concentration fields on the number of trajectories through each cell? How is the weighting performed, exactly? Does it only reduce noise in the spatial fields, as stated, or does it also affect their magnitude?**

AR11. It only reduces noise. To show this we added a figure in the Supplement (Fig. S7):

[Figure]

**Figure S7:** Same as Fig. 13 in the main paper, but without any weighting applied to the backtrajectory cells. (a) Output of the Concentration Field TSM, using the sum of the contributions from PMF nitrate- and sulfate-rich modes as concentration variable at the receptor (Aosta–Downtown). (b) CF based on the traffic and heating mode from PMF. Trajectories are cut at the borders of the COSMO-I2 domain.

The new figure (Fig. S7) was obtained without any weighting. No substantial differences can be noticed compared to Fig. 13.

[Figure]

**Figure 13:** (a) Output of the Concentration Field Trajectory Statistical Model, using the sum of the contributions from PMF nitrate- and sulfate-rich modes as concentration variable at the receptor (Aosta–Downtown). (b) Concentration Field based on the traffic and heating mode from PMF. Trajectories are cut at the borders of the COSMO-I2 domain.

We also tried to better describe the weighting procedure in the revised Sect. 3.2:

*To reduce statistical noise, every value $P_{ij}$ of the resulting map was then multiplied by a weighting factor, $w_{ij}$, linearly varying from 0 (for $N_{ij} \leq 20$ end points in a cell) to 1 (for $N_{ij} \geq 200$), i.e. $w_{ij} = min\{1, \frac{max\{0, N_{ij}-20\}}{180}\}$. To provide an idea of the effect of this weighting procedure, TSM maps without any weighting (i.e., $w_{ij} = 1$) have been included in the Supplement (Fig. S7).*

**RC12. Page 10, line 20. "...was chosen based on the Q/Qexp ratio". Please clarify.**

AR12. Section 3.3 now includes a short explanation of *the $Q/Q_{exp}$ ratio. The latter is the ratio between the objective function obtained with the selected number of factors ($Q$, introduced before) and its expected value ($Q_{exp}$). Elevated (i.e., $> 2$) $Q/Q_{exp}$ ratios could indicate that some samples and/or species are not well modelled and could be better explained by adding another source (Norris and Duvall, 2014).*

**RC13. Page 13, table 2. It seems to me that the criteria outlined here are not mutually exclusive. For instance, would a 13-hour period with wind speed exceeding 4 m/s be classified as "downwind" or as "foehn"? I also have a few concerns about the terminology.**
**(A) "Upwind" and "downwind" typically identify position relative to the wind and to a specific location (e.g. "when the flow is southerly, the city is downwind of the mountains"). In this context, it might be more appropriate to speak of "Channeled synoptic flow from the west/east". In addition, it would be appropriate to specify wind direction ranges for the two classes.**
**(B) "Breezes" have by definition wind speed > 1.5 m/s (Beaufort scale). This would exclude wind calms at night . It might be better to speak of "Diurnal wind system".**
**(C) "Stability" refers to the thermal structure of the atmosphere. Because winds are emphasized here, it might be appropriate to speak of "Wind calm".**
**(D) The directional range for the Foehn class probably deserves being explained.**
**(E) Please use wind "direction" instead of "provenance". "Direction" refers conventionally to the direction the wind comes from.**

AR13. Thanks for the remark, we acknowledge that the description of the criteria was not clear enough in the original manuscript. Actually, in our classification code the two classes (formerly called "Downwind" and "Foehn") were and are mutually exclusive. For example, in the case presented by the reviewer, the episode would be classified as "Foehn", since this category takes precedence over the synoptic winds. We now better specify this point in the updated Table 2 (below).

(A), (B) (C) and (E): the terminology suggested by the reviewer is now used throughout the revised manuscript (both the text and the figures have been updated).

(B): it must be noticed that nighttime and daytime conditions must be met for the same day. This is done in order to discriminate the "Diurnal wind systems" (inactive at night) from synoptic winds. A brace and a footnote in the revised Table 2 should make it clear, now.

(D): as explained in a new footnote, *the directional range for the Foehn cases was determined experimentally by comparison with manual classification from a trained weather observer.*

The changes to Table 2 are shown hereafter:

**Table 2.** Classification of weather regimes used in the study, based on wind speed ($|\boldsymbol{v}|$), wind direction, daily duration of the condition ($\Delta t$) and other measured meteorological variables.

| Primary classification | Secondary classification | Definition |
|---|---|---|
| Easterly winds | Channelled synoptic flow from the east | $|\boldsymbol{v}| > 1$ m s$^{-1}$, $\Delta t > 12$ h, wind direction 0–180 ° |
| | Diurnal wind systems (east) | Night (19–8 UTC)[a]: calm wind ($|\boldsymbol{v}| < 1$ m s$^{-1}$) or low westerly wind ($|\boldsymbol{v}| < 1.5$ m s$^{-1}$) |
| | | Day (8–19 UTC)[a]: easterly wind ($|\boldsymbol{v}| > 1.5$ m s$^{-1}$), $\Delta t > 4$ h |
| Westerly winds | Foehn (west) | $|\boldsymbol{v}| > 4$ m s$^{-1}$, $\Delta t > 4$ h, wind direction $< 25°$ or $> 225°$ [b], RH$< 40\%$ |
| | Channelled synoptic flow from the west | no Foehn, $|\boldsymbol{v}| > 1$ m s$^{-1}$, $\Delta t > 12$ h, wind direction 180–360 ° |
| Wind calm | – | $|\boldsymbol{v}| < 1$ m s$^{-1}$, $\Delta t > 20$ h |
| Other | Precipitation | Accumulated daily precipitation $> 1$ mm, $\Delta t > 4$ h |
| | Unclassified | Every day that does not fall in the previous categories |

[a] Both conditions must be met in order to discriminate such diurnal wind systems (inactive at night) from synoptic winds.

[b] The directional range for the Foehn cases was determined experimentally by comparison with manual classification from a trained weather observer.

**RC14. Page 14, figure 4. In summer, about 70% of the days feature breeze systems, but aerosol-layer days are only about 50% (Figure 3). What about the rest? Are easterly breezes occasionally "clean"?**

AR14. A comment about this (only apparent) discrepancy has been added to Sect. 4.1:

*For cases of easterly winds and no aerosol layer found (7%), possible reasons are:*

- *the diurnal wind, although clearly detected by the surface network, was of too short duration or too weak to transport the polluted air masses from the Po basin to Aosta–Saint-Christophe (at least ∼ 40 km must be travelled by the air masses along the main valley). In our record, this was for example the case of 14 and 18 September 2015, and 17 June 2016;*

- *back-trajectories were indeed channelled along the central valley, however they did not come from the Po basin, but rather from the other side of the Alps. This was the case, for instance, of 20 August 2015, in which air masses came from the Divedro Valley (north-east of the Aosta Valley) after crossing the Simplon pass (between Switzerland and Italy).*

*These exceptions also help understand why, in summer, about 70% of the days feature breeze systems (Fig. 4) while aerosol layers are detected only about 50% of the days (Fig. 3). Indeed, some part (7%, Fig. 5a) of this 20% discrepancy can be attributed to the above events, the remaining 13% being likely associated with days with a complex aerosol structure (not classified in Fig. 2) or to days affected by low clouds and/or desert dust events. These days were therefore labelled as "Other" in Fig. 3.*

**RC15. Page 15, line 14. It would be useful to explain where the Divedro Valley is. The great majority of readers wouldn't have a clue otherwise.**

AR15. Done, the updated text is cited in the previous reply (AR14).

**RC16. Page 16, line 15. Please remove the comma between "simple" and "forecasting".**

AR16. Done.

**RC17. Page 19, line 4. Reference to Teixeira et al 2016. The interested reader wouldn't find much in this editorial. More appropriate references could be those available at these DOIs: 10.3389/feart.2015.00077 and 10.3390/atmos9030102.**

AR17. The bibliographic reference has been changed according to the reviewer's comment.

**RC18. Page 25, line 5. The dashed continuous line in Figure 12a is very hard to see. Consider replacing with a colored line, or with a filled area in the background.**

AR18. Thank you for this suggestion. The dashed line has been replaced with a filled area. The new figure is presented below:

[Figure]

**Figure 12:** (a) Temporal chart of the contribution of secondary (nitrate-rich and sulfate-rich) modes to the total PM$_{10}$. The shaded area represents the estimated local production of secondary aerosol. The difference between each dot and this baseline can thus be read as the non-local contribution to the aerosol concentration in Aosta–Downtown. Since ions chemical speciation started in 2017, only one year of overlap with the ALC is available at the moment (see Table 1). (b) Boxplot of the contribution of secondary (nitrate-rich and sulfate-rich) modes to the total PM$_{10}$ concentration as a function of the day type. PMF-dataset "a" was used for both plots.

**RC19. Page 25, Figure 13. Text is hard to read on the colored background. Consider using a different color map, or adding white background to the location labels.**

AR19. White background has been added to the location labels (refer to AR11 for the new figures).

**RC20. Pages 30-32. There is some redundancy in Figures 16-17-18-19. It is unclear why results from two locations are represented in the figure. In fact, the differences between the two sites are never discussed in detail in the text. It would be possible to remove the figure panels referring to Donnas without great loss of information.**

AR20. By including both sites we intended to show the effect of the distance from the Po Valley on the receptor site. However, given this comment, we have removed the figure panels referring to Donnas. Figure 17 has been updated as follows (also considering RC3 by Reviewer#1):

[Figure]

**Figure 17:** Absolute (a,c) and relative (b,d) differences between simulated and observed $PM_{10}$ concentrations at the surface in Aosta–Downtown. The mean bias error (MBE) and the normalised mean bias error (NMBE) for each case are reported in the plot titles. First row: FARM simulations as currently performed by ARPA. Second row: the $PM_{10}$ concentrations from outside the boundaries of the domain were multiplied by a factor $W=4$.

**RC21. Page 33, lines 2-3. "This is a clear indication that the external contribution (boundary conditions) is not optimally parametrised in the model".** I am really not sure that I agree on this statement. If I understand correctly, FARM uses information from the whole COSMO-I2 domain, which completely includes the Po valley. Therefore, the problem does not lie in "external contributions" not being represented in the lateral boundary conditions. There might be problems with the lower boundary conditions (emission inventories), but I doubt these can be wrong by a factor of 4 over the whole Po valley. I am not even sure that model parametrisations matter here.
I would argue that the relatively coarse resolution of the weather model (2.8 km, marginally enough to resolve the valley) and of the transport model (only 16 vertical levels) play an important role. Please consider the topography cross-sections in Figure 20. Aosta seems to lie in a basin, likely non-existent in reality and introduced in the model by terrain smoothing. If the topography profile descended continuously from Aosta to Donnas and the Po Plain (as in reality), it would be possible to resolve the horizontal advection of aerosol-laden air from much lower altitudes on the plain, likely removing a large fraction of the negative concentration bias.

AR21. First of all, it must be noticed that FARM does not use "information from the whole COSMO-I2 domain, which completely includes the Po valley". We tried to better explain this in Sect. 3.1:

*FARM is only run over a small domain (light blue rectangle in Fig. 1a,b), roughly corresponding to the Aosta Valley. A regional emission inventory ("local sources", updated to 2015) is supplied to the CTM over the same area to accurately assess the magnitude of the pollution load and its variability in time and space. Data from a national inventory and CTM model (QualeAria, here referred to as "boundary conditions", outer rectangle in Fig. 1a), taken along the border of the inner (light blue) rectangle, are also used to estimate the mass exchange from outside the borders of the FARM domain.*

As a second point we added further analysis to check whether FARM performs better in a less complex terrain. To this purpose, we decided to include the data from the Ivrea station. This addition clearly showed that the same $PM_{10}$ underestimation of the model is also obtained over flat terrain, demonstrating that this is not driven by inaccuracies of the simulated wind fields over the more complex terrain of Aosta. We think that this represents a remarkable improvement to our study, supporting the hypothesis that the model underestimations are mainly due to incorrect emission inventories/unaccounted physical processes (e.g., aqueous-phase chemistry) at a national level. This additional analysis required description of the measurement site, rationale, and comment of the results obtained (and relevant figures):

*In this study, we mainly used data from four sites (...) and Ivrea (243 m a.s.l., urban background) (...), a site in the Po Plain (31 $\mu g \, m^{-3}$ average $PM_{10}$ in 2017) located just outside the Aosta Valley, in the Italian Piedmont region (Fig. 1b). This was done to check if and how much inaccuracies of the model in reproducing the aerosol loads over the Aosta Valley are due to difficulties in simulating the wind field in such a complex terrain. In fact, the city centre of Ivrea (24000 inhabitants) is approximately in the middle between the measurement site (south of the city) and the nearest cell of our model domain (north of the city), the distance between these two points being only 2 km. The altitude of the cell (from the digital elevation model used in our simulations) is 241 m a.s.l., i.e. approximately the real one (...) The station of Ivrea features, among other instruments, a TCR Tecora Charlie/Sentinel $PM_{10}$ sampler. $PM_{10}$ concentrations are then determined by a gravimetric technique.*

We have remarkably expanded Sect. 4.5 by adding the dataset from this station to the study, and comparing it to the corresponding FARM simulations:

[revised manuscript text omitted]

Concerning the impact of smoothed topography, please refer to AR10.

**RC22. Page 33, line 10. "The reason for discrepancies is therefore likely in the emissions". Why not in deficiencies of the transport modelling, as I argued above?**

AR22. See AR21.

**RC23. Page 46, line 18: There seems to be a problem with the reference to Thunis et al, 2012 (the third author).**

AR23. Thank you, the reference has been corrected.